



Earth System
Science
Data

# Institute for Marine and Atmospheric Research Utrecht (IMAU) Antarctic automatic weather station data, including surface radiation balance (1995–2022)

**Maurice van Tiggelen, Paul C. J. P. Smeets, Carleen H. Reijmer, Peter Kuipers Munneke, and Michiel R. van den Broeke**

Institute for Marine and Atmospheric research Utrecht (IMAU), Utrecht University, Utrecht, the Netherlands

**Correspondence:** Maurice van Tiggelen (m.vantiggelen@uu.nl)

**Abstract.** In cooperation with multiple institutes, the Institute for Marine and Atmospheric Research Utrecht (IMAU) at Utrecht University has operated automated weather stations (AWSs) at 19 locations on the Antarctic ice sheet from 1995 through 2022. Besides standard meteorological measurements (pressure, temperature, humidity, wind speed, and direction), these stations include measured short-wave and long-wave radiation components and surface height, thereby allowing for the reliable in situ quantification of the surface energy balance (SEB) and surface mass balance (SMB) at hourly and 2 h temporal resolution. This unique dataset can be used for climate model evaluation and development, for the validation of remote sensing products, for the quantification of long-term climatological changes, for the interpretation of ice cores, and for process understanding in general. This paper describes the dataset and the applied measurement corrections. The total dataset contains 57 station years of data, of which 65 include both SEB and SMB observations, and is available at https://doi.org/10.1594/PANGAEA.974080 (Van Tiggelen et al., 2024).

## 1 Introduction

The Antarctic ice sheet is the largest reservoir of fresh water, holding a global sea-level potential of 58 m (Morlighem et al., 2020), and also acts as a reliable record of the recent climate (e.g. EPICA, 2004). Due to its isolation, dry climate, and long austral winter, it also provides unique and often favourable locations for meteorological, astronomical, geophysical, and upper-atmosphere observations.

Reliable in situ measurements of meteorological quantities and components of the surface energy budget (SEB) are required for climate model evaluation on the Antarctic ice sheet (e.g. Van Wessem et al., 2018; Souverijns et al., 2019; Mottram et al., 2021; Kittel et al., 2021), to provide realistic atmospheric forcing for snow/firn models (e.g. Wever et al., 2022; Le Moigne et al., 2022), and to validate satellite remote sensing products (e.g. Trusel et al., 2013), but also to train machine learning algorithms (e.g. Hu et al., 2021), to interpret ice core observations (e.g. Laepple et al., 2011),

and to provide in situ estimates of the sub-seasonal surface mass balance (SMB; e.g. Reijmer and van den Broeke, 2003). In addition, such in situ observations serve as a basis for process understanding, such as surface sublimation (Thiery et al., 2012; Amory, 2020), surface melt quantification (Jakobs et al., 2020), formation and burial of meltwater lakes (Buzzard et al., 2018; Dunmire et al., 2020), blue ice formation (Bintanja and Reijmer, 2001), the formation of impermeable ice slabs (Kuipers Munneke et al., 2018), and the detection of climate trends (e.g. Turner et al., 2016).

Despite the increasing need for reliable in situ measurements, automated weather station (AWS) observations are still scarce in Antarctica. The first continuous AWS observations started in 1978 (Lazzara et al., 2012), and since then, at most 146 AWSs were simultaneously recording near-surface meteorological variables across the continent (Wang et al., 2023). These AWSs were part of one of several networks, such as the Antarctic Meteorological Research Center (AMRC) network maintained by the University of

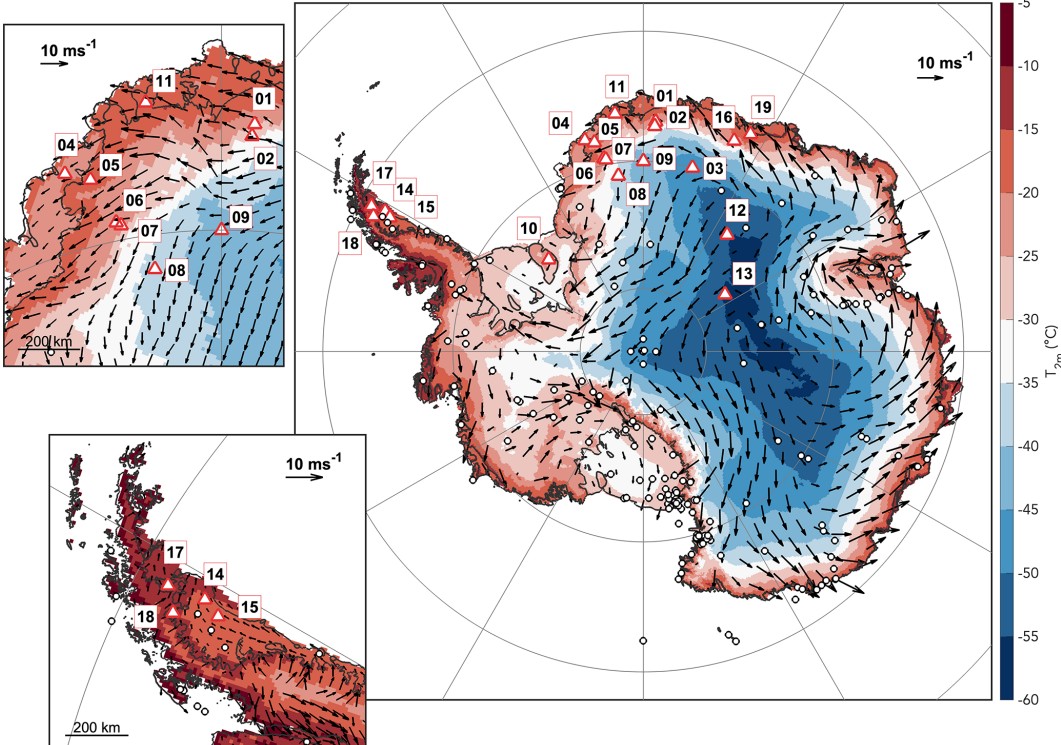

**Figure 1.** Location of the AWS presented in this database (red triangles). Background colour denotes the modelled annual average 2 m near-surface air temperature from the RACMO2.4p1 regional climate model during the period 1990–2020 (van Dalum et al., 2024). Black circles denote AWS from the AntAWS database (Wang et al., 2023). Average 10 m near-surface wind vectors from RACMO2.4p1 are also shown. Insets are shown for Dronning Maud Land (top left panel) and the Antarctic Peninsula (bottom left panel).

Wisconsin-Madison (Lazzara et al., 2012), the Australian Antarctic Division (AAD) network (Allison, 1998), the Italian National Program of Antarctic Research (PNRA) network (Grigioni et al., 2016), the Chinese National Antarctic Research Expedition (CHINARE) PANDA network (Ding et al., 2022), the British Antarctic Survey (BAS) network, the Japanese Antarctic Research Expedition (JARE) network (Kurita et al., 2024), the French Antarctic Program (Institut Polaire Francais-Paul Emile Victor, IPEV) network, and other similar networks maintained by different nations or organizations. These stations are shown in Fig. 1 and further described in Wang et al. (2023). The number of AWSs recording the near-surface radiation balance and surface height change, allowing for the estimation of the SEB and SMB, is even much more limited. Since 1995, the Institute for Marine and Atmospheric Research Utrecht (IMAU), in cooperation with the Alfred Wegener Institute (AWI), the BAS, the Finnish Antarctic Research Program (FINNARP), the Norwegian Polar Institute (NPI), the United States Antarctic Research Program (USARP), the Swedish Antarctic Research Program (SWEDARP), the Royal Meteorological Institute of Belgium (KMI), and the KU Leuven, has been operating 19 AWSs in Antarctica, of which 16 measured the radiation components, allowing for a direct quantification of

the entire SEB. Recently, Jakobs et al. (2020) presented a database containing SEB calculations at eight stations from the IMAU network where surface melt occurs.

In this work we describe the dataset of measurements from all these 19 AWSs that spans from 1995 to 2022. We describe the applied corrections and processing steps and also provide a single quality flag as well as the interpolated wind speed, air temperature, and specific humidity at standard heights of 10 and 2 m above the surface using similarity theory, which allows for an easier comparison with different datasets. Although the measurements from this dataset are also partly contained in the datasets from Jakobs et al. (2020) and Wang et al. (2021), the data presented in this work have gone through an elaborate quality control and data correction strategy, which are specifically tailored for the unique combination of sensors and locations of the IMAU dataset. On the other hand, this dataset does not contain the calculation of SEB/SMB fluxes, which requires additional model calculations.

**Table 1.** Metadata for all the IMAU AWSs in Antarctica. For station types I and II, the locations are taken from fieldwork reports at the installation date, while for station type III the locations are from the recorded GPS position at the installation date. The surface elevation is taken from the REMA DEM (Howat et al., 2022) as the average values within 500 m distance from each AWS. As of December 2022, AWS14 was still operational, and AWS18 was moved 23 km and renamed to AWS20. Note that AWS20 is not part of this dataset.

| Station name | Location | Latitude (° N) | Longitude (° E) | Elevation (m a.s.l) | Operation period (start–end) | AWS type | Cooperating institute |
|---|---|---|---|---|---|---|---|
| AWS01 | NARE9697 Site A | −71.900 | 3.083 | 1472 | 1 Jan 1997–12 Dec 2000 | I | NPI |
| AWS02 | NARE9697 Site C | −72.251 | 2.891 | 2419 | 12 Feb 1997–13 Dec 2000 | I | NPI |
| AWS03 | NARE9697 Site M | −75.000 | 15.002 | 3470 | 28 Jan 1997–14 Jan 2001 | I | NPI |
| AWS04 | Rampen site 1090 | −72.753 | −15.499 | 59 | 19 Dec 1997–29 Dec 2002 | II | SWEDARP |
| AWS05* | Wasa/Aboa Camp Maudheimvida | −73.105 | −13.165 | 366 | 2 Feb 1998–7 Feb 2014 | II | FINNARP/SWEDARP |
| AWS06* | Svea Cross | −74.482 | −11.517 | 1100 | 14 Jan 1998–16 Feb 2009 | II | SWEDARP |
| AWS07 | Scharffenbergbotnen (blue ice) | −74.578 | −11.048 | 1175 | 31 Dec 1997–6 Jan 2003 | II | SWEDARP |
| AWS08 | Camp Victoria | −76.000 | −8.050 | 2398 | 12 Jan 1998–7 Jan 2003 | II | SWEDARP |
| AWS09* | Kohnen Station/EPICA DML05 | −75.003 | 0.007 | 2892 | 29 Dec 1997–18 Nov 2022 | II | AWI |
| AWS10 | Thyssen Höhe, Berkner Island | −79.567 | −45.782 | 867 | 12 Feb 1995–18 Jul 2005 | I/II | AWI/BAS |
| AWS11 | Halvfarryggen ice rise | −71.175 | −6.800 | 700 | 13 Feb 2007–31 Jan 2019 | II/III | AWI |
| AWS12* | Plateau Station B | −78.650 | 35.633 | 362 | 15 Dec 2007–10 Mar 2016 | II | NPI/USARP |
| AWS13* | Pole of inaccessibility | −82.117 | 55.033 | 3723 | 2 Jan 2008–11 Mar 2016 | II | NPI/USARP |
| AWS14* | Larsen C North | −67.013 | −61.480 | 47 | 21 Jan 2009–still running | II/III | BAS |
| AWS15* | Larsen C South | −67.572 | −62.125 | 49 | 21 Jan 2009–6 May 2014 | II | BAS |
| AWS16 | Princess Elisabeth station | −71.949 | 23.358 | 1371 | 2 Feb 2009–3 Jul 2020 | II/III | KU Leuven/KMI |
| AWS17* | Scar Inlet, Larsen B remnants | −65.933 | −61.850 | 72 | 19 Feb 2011–10 Mar 2016 | II/III | BAS |
| AWS18 | Cabinet Inlet, Larsen C West | −66.402 | −63.371 | 78 | 25 Nov 2014–2 Dec 2022 | III | BAS |
| AWS19 | King Baudouin ice shelf | −70.963 | 26.255 | 60 | 10 Dec 2014–2 Feb 2016 | III | KU Leuven/KMI |
| AWS20 | Cabinet Inlet, Larsen C West | −66.616 | −63.229 | 70 | 3 Dec 2022–still running | n/a | BAS |

\* Marks the stations from which uncorrected data were transmitted to the global telecommunication system (GTS) for a period of time: AWS05 from 2 February 2003 until end; AWS06 from 9 January 2003 until end; AWS09 from 10 January 2001 until end; AWS12, AWS13, and AWS15 for the full period; AWS14 and AWS17 from the start through 18 January 2015.
n/a: not applicable.

## 2 AWS data description

### 2.1 Location and installation

The location, name, and period of operation of each AWS are given in Fig. 1 and Table 1. The first station (AWS10) was erected in February 1995 on Thyssen Höhe, the south dome of Berkner Island, in cooperation with the AWI and the BAS in support of palaeoclimate reconstructions from deep ice coring (Reijmer et al., 1999). This AWS provided the climatological background for the medium deep ice core drilling project in 1994/1995 (AWI) and the deep drilling project in 2003/2005 (BAS). During the austral summer of 1996/1997, three AWSs (AWS01, AWS02 AWS03) were installed in central Dronning Maud Land (DML) in collaboration with the NPI during the Norwegian/Swedish/-Dutch NARE9697 ground traverse (Winther et al., 1997; Van den Broeke et al., 1999), as part of the European Project for Ice Coring in Antarctica (EPICA) DML pre-site survey. In the austral summer of 1997/1998, five additional AWSs (AWS04–AWS08) were installed in DML in collaboration with the SWEDARP during the Swedish/Norwegian/-Dutch ground traverse SWEDARP9798 (Holmlund et al., 2000), and AWS09 was installed by AWI close to the EPICA DML drilling site at the Kohnen AWI station (Reijmer and van den Broeke, 2003). All these stations were part of the Netherlands' contribution to EPICA (EPICA, 2006).

As part of the International Polar Year (IPY 2007–2008), AWS11 was erected in January 2007 near the top of the Halvfarryggen ice rise by AWI (Drews et al., 2013). Halvfarryggen is located about 80 km from Neumayer III station, and the AWS was installed in support of a coastal deep ice core drilling project. During the austral summer of 2007/2008, one AWS was installed at former Plateau Station B (AWS12) and one AWS at the Pole of Inaccessibility (AWS13) along the Norwegian–US scientific traverse of East Antarctica (Goldman, 2008), in collaboration with NPI and the Cooperative Institute for Research in the Environmental Sciences (CIRES). These two stations provided the climatological background in search of a favourable new deep drilling site in interior East Antarctica for drilling the oldest ice (Van Liefferinge et al., 2018). In 2009, two stations (AWS14, AWS15) were erected on the Larsen C ice shelf in collaboration with BAS, CIRES, and the Jet Propulsion Laboratory (Kuipers Munneke et al., 2012), followed by the installation of AWS17 in February 2011 at Scar Inlet, on the remnants of the Larsen B ice shelf, and the installation of AWS18 in December 2014 in Cabinet Inlet, near the grounding line of Larsen C ice shelf (Kuipers Munneke et al., 2018).

In collaboration with the KMI and KU Leuven, AWS16 was installed in DML at the Belgian Princess Elizabeth station in February 2009 (Gorodetskaya et al., 2013), and AWS19 was installed in December 2014 near the grounding

line of Roi Baudouin ice shelf, 150 km from Princess Elizabeth station (Lenaerts et al., 2017).

At present, only AWS14 is still operational, and in December 2022, AWS18 was reinstalled 23 km away from the grounding line due to melt pond formation complicating the yearly maintenance visits and renamed as AWS20, which is not part of this dataset. Maintenance visits were in general performed using a standard procedure contained in a form describing a list of actions (i.e. make photographs at arrival, note anything unusual, measure yard directions and heights at arrival and departure, check data logger data, and replace the memory module). Sensors were commissioned to be replaced on a regular basis. Additional instructions and replacements were added in the case of transmitted ARGOS data indicating failure of a sensor, of the data logger, or of the power supply. AWS14 and AWS18 are part of this dataset and were maintained by IMAU in collaboration with BAS on the Larsen C ice shelf, where they experience an average lateral displacement of 451 and 193 m yr$^{-1}$, respectively, due to ice flow. The other stations were removed, either because it was anticipated that these sites would not be visited again in the near future (AWS01–03, 06, 08, 10) or because they were funded for short project periods (AWS11, 12, 13, 14, 17). AWS04 was removed due to high accumulation rates necessitating frequent excavation, and AWS07 was removed due to frequent damage by strong winds. The continuation of AWS measurements at AWS05 was taken over by the University of Helsinki and FINNARP, and the continuation at AWS09 was taken over by AWI.

## 2.2  AWS design

Since 1995, three different types of AWS designs were used. Each station consists of a four-legged frame with an extensible vertical mast consolidated by guy wires and a horizontal boom on which the sensors are mounted (Fig. 2). The initial installation height of the sensor boom above the surface varies across sites and between maintenance visits and ranges between 2.6 and 5 m. All stations measure typical meteorological parameters such as air temperature, wind speed, wind direction, downward short-wave radiation, air pressure, snow temperature, and surface height but differ in sensor specification and placement, power supply, and sampling strategy. Station types II and III were also upgraded with sensors that allow for the estimation of all the SEB components. The sensor specifications per AWS type are given in Table 2. The type I stations (AWS01, AWS02, AWS03, and AWS10), installed during the austral summers of 1995–1996 and 1996–1997, mainly used Aanderaa sensors including a cup anemometer in combination with a wind direction vane and did not record air humidity, only the incoming short-wave component of the net surface radiation (Fig. 2a). The type II station was used at most locations from austral summer 1997/1998 until austral summer 2014/2015 (Fig. 2b) and mainly used Vaisala sensors for air temper-

ature, humidity, and pressure; a R. M. Young wind vane for wind speed and direction; a Kipp & Zonen CNR1 radiometer for the four radiation components; and a Campbell SR50 sonic height ranger for surface height. After 2015, a more compact and lower-power design was used at AWS11, AWS14, AWS16, AWS17, AWS18, and AWS19 (type III; Fig. 2c), which is also referred to as an intelligent weather station (iWS). These stations also use a R. M. Young wind vane and a Kipp & Zonen CNR1 or CNR4, but temperature, humidity, pressure, and the surface height sensors were replaced by custom-assembled sensors in a single compact housing. At all locations, the AWSs were also fitted with thermistor strings to measure the subsurface temperatures. However, these subsurface data are not part of this quality-controlled dataset, since the exact installation depth of the subsurface sensors is not known for all maintenance visits.

The stations sample every 5 (type I) or 6 min (types II and III), after which 2 h (type II), 1 h (types I and II), or half-hour (type III) averages are calculated, stored locally, and transmitted using Argos transmitters. The stations are powered by either lithium or alkaline batteries, in combination with a solar panel for all type III stations and for the type II station at AWS09. The sensors are neither actively ventilated nor heated to minimize power consumption and to ensure their continuous operation when left unattended for long periods of time. For a period of their operation, measurements of stations 5, 6, 9, 12, 13, 14, 15, and 17 have been transmitted to the global telecommunication system (GTS). The exact periods are given in Table 1.

## 2.3  Sensor corrections

In the following we describe the corrections applied to the dataset. Some of these are also partly described for Greenland IMAU AWS data by Smeets et al. (2018) but have been adapted for the specific sensors and design used in this dataset. The corrections are applied separately for each station and each period of data available in between maintenance visits to accommodate the replacement or repair of sensors and changes in sensor orientation and heights. All unheated meteorological instruments operating under polar conditions may suffer from riming or hoar frost deposition, which we assume to occur when the relative humidity exceeds 90 %, the air temperature is lower than 0 °C, and the absolute value of the net long-wave radiation is smaller than 2 W m$^{-2}$ for at least 24 consecutive hours. These observations are flagged but are not removed from the dataset.

### 2.3.1  Temperature correction

The heating of the temperature sensor in the passively ventilated radiation screens causes an excess temperature, which peaks during conditions with high insolation and low wind speeds. The correction is empirically determined per station type by comparing with measurements from a separate

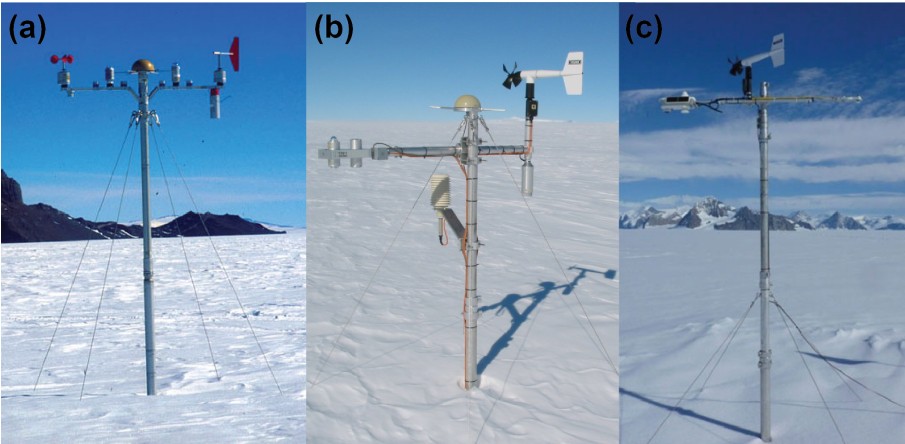

**Figure 2.** Photographs of the three different station types taken during maintenance visits: type I at AWS01 **(a)**, type II at AWS06 **(b)**, and type III at AWS18 **(c)**.

**Table 2.** Instruments used for the different types of IMAU AWSs. The years denote the approximate period of operation per AWS type.

| AWS | Variable | Instrument or sensor | Range | Accuracy[a] |
|---|---|---|---|---|
| Type I 1995–1997 | Air temperature | Aanderaa 2775C | −90 to +30 °C | 0.1 °C at −20 °C |
| | Air pressure | Aanderaa 2775C | 600 to 1024 hPa | 0.5 hPa |
| | Wind speed | Aanderaa 2740 | 0.5 to 76 m s$^{-1}$ | 0.5 m s$^{-1}$ |
| | Wind direction | Aanderaa 2750 | 0 to 360° | 5° |
| | Short-wave radiation | Aanderaa 2770 | 300–2500 nm, 0 to 2000 W m$^{-2}$ | < 20 W m$^{-2}$ |
| | Snow temperature | Aanderaa | −70 to +30 °C | 0.1 °C |
| | Surface height | Aanderaa | 1 to 4 m | 0.01 m |
| | Data logger | Campbell CR10 | – | – |
| Type II 1998–2014 | Air temperature | Vaisala HMP35AC | −80 to +56 °C | 0.3 °C |
| | Relative humidity | Vaisala HMP35AC | 0 % to 100 % | 2 % (RH < 90 %) |
| | Air pressure | Vaisala PTB101B | 600 to 1060 hPa | 0.5 hPa |
| | Wind speed | R. M. Young 05103 | 0 to 60 m s$^{-1}$ | 0.3 m s$^{-1}$ |
| | Wind direction | R. M. Young 05103 | 0 to 360° | 3° |
| | Short-wave radiation | Kipp & Zonen CNR1 | 305–2800 nm | 10 % daily totals |
| | Long-wave radiation | Kipp & Zonen CNR1 | 5–50 μm | 10 % daily totals |
| | Snow temperature | Vaisala HMP35AC | −80 to +56 °C | 0.3 °C |
| | Surface height | Campbell SR50 | 0.5 to 10 m | 0.01 m or 0.4 % |
| | Air temperature 2[b] | Campbell PT100 | 0.5 to 10 m | 0.1 °C |
| | Data logger | Campbell CR10 | – | – |
| Type III 2015–2023 | Air temperature | NTC thermistor | −60 to 40 °C | 0.1 °C |
| | Relative humidity | Sensirion SHT25 | 0 % to 100 % | 1.8 % |
| | Air pressure | Freescale Xtrinsic MPL 3115A2 | 200 to 1100 hPa | 4 hPa |
| | Wind speed | R. M. Young 05103 | 0 to 60 m s$^{-1}$ | 0.3 m s$^{-1}$ |
| | Wind direction | R. M. Young 05103 | 0 to 360° | 3° |
| | Short-wave radiation | Kipp & Zonen CNR4 | 300–2800 nm | < 5 % daily totals |
| | Long-wave radiation | Kipp & Zonen CNR4 | 4.2–42 μm | 10 % daily totals |
| | Snow temperature | PS222J2 | −80 to +56 °C | 0.1 °C |
| | Surface height | MaxBotix HRXL-MaxSonar-WRS | 0.5 to 5 m | 0.01 m or 0.4 % |
| | Tilt | HMC6343 | −179.9 to 179.9° | 0.1° |
| | Data logger | Custom-made at IMAU | – | – |

[a] Reported accuracy by the manufacturer. [b] The PT100 air temperature sensor is only installed at AWS09 after January 2008 and at AWS12 and AWS13.

https://doi.org/10.5194/essd-17-1-2025

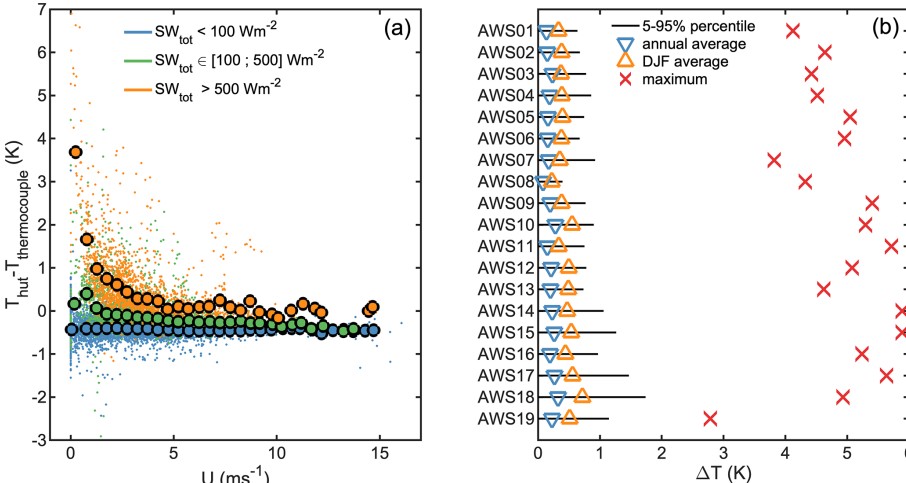

**Figure 3. (a)** Example of temperature excess versus horizontal wind speed for three different classes of total short-wave radiation ($SW_{tot} = SW_d + SW_u$) at AWS14 in the period December 2019–October 2020 when an additional thermocouple measurement was available. Note that a bias of about $-0.4\,°C$ is still present in the thermocouple readings. **(b)** Annual average (blue triangles), DJF average (orange triangles), annual maximum (red crosses), and 5 %–95 % percentile range (black lines) of the correction for solar heating of the temperature in the radiation shield during the entire measurement period per station.

thin wire thermocouple corrected for radiation errors, following Jacobs and McNaughton (1994) and Smeets et al. (2018). For station types I and II, we use simultaneous measurements of a 0.003 inch (76.2 μm) fine-wire thermocouple (Campbell FW3 Type E) and HMP35 probe (Vaisala) inside a Young multi-plate radiation shield from August 2003 to August 2004 at location S6 on the Greenland ice sheet (Smeets et al., 2018) to fit the excess temperature to the following empirical relation:

$$\Delta T = \frac{SW_{tot}}{A(12U)^{1.3}}, \tag{1}$$

with $\Delta T$ the excess temperature correction that is subtracted from the raw measurements, $SW_{tot} = SW_d + SW_u$ the total measured short-wave radiation, $SW_d/SW_u$ the downward/upward short-wave radiation, $A$ a geometry factor that depends on the sensor and radiation shield, and $U$ the horizontal wind speed at sensor height. For AWS type II, $A = (9.59 \times 10^{-3} SW_{net} + 6.32)/17$, which is determined empirically based on data from S6. We also use this relation for station type I since no information on radiation shield characteristics is available. At very low wind speeds ($U < 2\,\mathrm{m\,s^{-1}}$), we estimate $\Delta T$ from interpolating between points $\Delta T(U = 0)$ and $\Delta T(U = 2)$, with $\Delta T(U = 0) = k \times 4.14 \times 10^{-3} - 0.15$. An example of the temperature excess is shown in Fig. 3a, in which the uncorrected temperature measured in the radiation shield is compared to uncorrected thermocouple measurements for various wind speeds and values of total short-wave radiation. The accuracy of the temperature excess correction is deemed better than $1\,°C$ for wind speeds exceeding $1\,\mathrm{m\,s^{-1}}$.

Type III stations have different radiation shields than previous types and have sufficient reference thermocouple measurements available that allow for a different, more automated correction procedure. The procedure is based on Nakamura and Mahrt (2005) and Huwald et al. (2009). The temperature excess is fitted to the following non-dimensional quantity:

$$C = 1000 \frac{SW_{tot}}{\rho_a C_p U T}, \tag{2}$$

where $\rho_a$ is the air density and $C_p$ the air specific heat capacity at constant pressure. The temperature excess is fitted to the non-dimensional quantity $C$ with a first-order and a third-order polynomial using measurements from an additional thermocouple corrected for heating, if available, or using the results from multiple other stations equipped with thermocouples and the same radiation shield. The first-order polynomial is used for large $\Delta T$ values in order to prevent an unrealistic correction for very low wind speeds and high insolation (i.e. $C < 8$).

On average, this correction reduces the measured air temperature between 0 to $1\,K$ at all AWSs but may reach up to $6\,K$ during brief periods with high insolation, low wind speeds, and less effective radiation shields (Fig. 3b).

### 2.3.2    Relative humidity

The hygrometers measure relative humidity with respect to liquid water; hence their output needs to be rescaled to relative humidity with respect to ice. Furthermore, we rescale the humidity values for temperatures below $0\,°C$ such that the

highest RH values, deemed close to saturation, are also close to 100 % measured relative humidity. We employ a similar method to that described by Anderson (1994). At the end of the correction procedure, we also correct for the fact that the relative humidity was measured inside an excess heated radiation shield instead of the ambient air.

The procedure we employ is as follows. First, we bin the raw, uncorrected RH values in 1 °C windows using the raw, uncorrected air temperature, and we keep the highest 95th-percentile values per bin if at least 20 values are present. Then, we first try a second-order fit for all data up to −4 °C, and if the curvature of the fit is positive, i.e. when the upper bound of RH values increases for very low temperatures, we use a linear fit instead. Additionally, for stations where the minimum air temperature bin lies below −60 °C, we determine a second linear fit for all air temperature values up to −55 °C. The latter allows for some overlap between the two fits around −60 °C. The reason for the different fits is the varying sensor characteristics over the vast temperature range encountered within the dataset, ranging from the coastal stations up to the Antarctic plateau. The polynomials are then rescaled using the RH offset at 0 °C compared to 100 %. The resulting functions determine the upper bound of the raw RH measurements versus air temperature, which are used to select data found within ±1 % of these curves. We recalculate the raw RH values into their values over ice and then fit a fourth-order polynomial that is used to rescale the RH value to 100 % over the entire temperature range, as is done in Anderson (1994).

The final step is the correction due to different saturated water vapour pressures in the radiation shield and in the ambient air (e.g. Makkonen and Laakso, 2005), which is only applied to RH values lower than 98 % in order to prevent the correction to result in RH values far above 100 %.

### 2.3.3 Pyrgeometer (long-wave radiation)

If not affected by rime of frost, readings from an unventilated pyrgeometer are affected by (1) window heating due to absorption of solar radiation, (2) instrumental biases, and (3) the emission of long-wave radiation in the air column between the surface and the sensor under conditions of strong vertical temperature gradients.

First, we correct for window heating in the following empirical way, based on measurements taken during an intercomparison experiment at the Baseline Surface Radiation Network (BSRN) site of the Royal Netherlands Meteorological Institute (KNMI) located at Cabauw, in the Netherlands (Knap, 2022). The results illustrated a dependence of window heating on sensor type (CNR1 or CNR4), short-wave radiation heating the window, and wind speed cooling the window, which is also confirmed by stations presented in this dataset (Fig. 4a). The window heating effect can be described as a ratio between incoming short-wave radiation and wind speed:

$$\Delta \mathrm{LW}_{\mathrm{u,d}} = a \frac{\mathrm{SW}_{\mathrm{u,d}}}{U^b}, \tag{3}$$

with $a$ a sensitivity coefficient for window heating that depends on the sensor type (0.025 for model CNR1 and 0.0125 for model CNR4), $b = 0.35$ a sensitivity coefficient for wind speed cooling, and $\Delta \mathrm{LW}_{\mathrm{u,d}}$ the window heating correction that is computed for both long-wave components separately and subtracted from the raw measurements.

Then, we correct for instrumental biases using the following procedure. For each station we select measurements with $\mathrm{SW}_{\mathrm{d}}$ lower than $50\,\mathrm{W\,m^{-2}}$ to rule out any influence from window heating. We use a twofold method depending on data availability within the temperature range. The correction is illustrated in Fig. 4b.

For stations where there are no remaining data with $T > 0\,°\mathrm{C}$, we further only select the data during near-neutral conditions to minimize the influence of temperature effects. Near-neutral conditions are defined as when wind speeds are higher than $6\,\mathrm{m\,s^{-1}}$, relative humidity above 80 %, temperatures below −10 °C, and a temperature difference between the ambient air and the radiometer body smaller than 0.5 °C. We define the bias as the median of the remaining $\mathrm{LW}_{\mathrm{u}} - \sigma T^4$ data, with $T$ the air temperature and $\sigma = 5.67 \times 10^{-8}\,\mathrm{W\,m^{-2}\,K^{-4}}$ the Stefan–Boltzmann constant. We assume that $\mathrm{LW}_{\mathrm{d}}$ has the same offset, so the bias is then subtracted from the entire dataset for both long-wave components.

For stations with remaining data with $T > 0\,°\mathrm{C}$, we bin all the $\mathrm{LW}_{\mathrm{u}}$ data for $\mathrm{SW}_{\mathrm{d}} < 50\,\mathrm{W\,m^{-2}}$ as a function of air temperature in windows of 0.2 °C and compute the median of the five largest values within each bin. We then compute the linear regression of these maxima versus air temperature and define the bias as the intercept of the linear fit for $T = 0\,°\mathrm{C}$. We interpret this bias as instrumental error; hence we subtract it from the entire dataset for both long-wave components. In addition, we interpret the remaining linear offset for $T > 0\,°\mathrm{C}$ as long-wave radiation divergence; hence we also remove/add this linear dependency from the measured $\mathrm{LW}_{\mathrm{u}}/\mathrm{LW}_{\mathrm{d}}$, respectively, in order to obtain measured long-wave radiation components at the surface.

When averaged over the entire time series per station, and including both the correction for window heating, instrumental bias, and long-wave radiation divergence, the downward long-wave radiation component is reduced by between 0 and $10\,\mathrm{W\,m^{-2}}$ (Fig. 4c). The largest correction of $40\,\mathrm{W\,m^{-2}}$ is found at AWS17 during a co-occurring period with both high air temperature and large insolation. Type III stations have the largest correction, partly due to their geographical location on ice shelves, but also due to a different instrumental offset compared to type II.

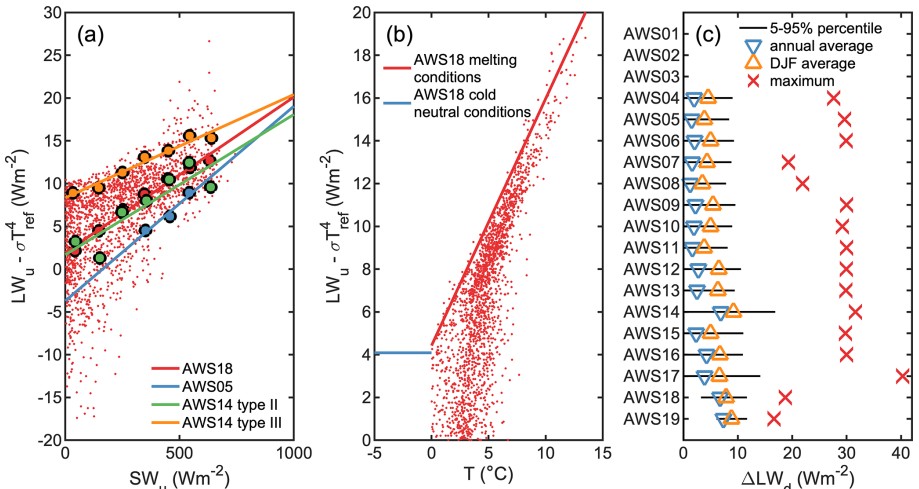

**Figure 4. (a)** Pyrgeometer window heating: excess upward long-wave radiation versus upward short-wave radiation and corresponding bin averages and linear regressions for daytime data when solar zenith angles are below 90° and when air temperatures are in the range of 2–5 °C. **(b)** Sensor bias and emitted long-wave radiation by the air: excess upward long-wave radiation versus air temperature for nighttime melting condition data (red) and nighttime cold neutral conditions (blue) at AWS18 and corresponding linear regressions. **(c)** Annual average (blue triangles), DJF average (orange triangles), annual maximum (red crosses), and 5 %–95 % percentile range (black lines) of the correction for downward long-wave radiation during the entire measurement period per station.

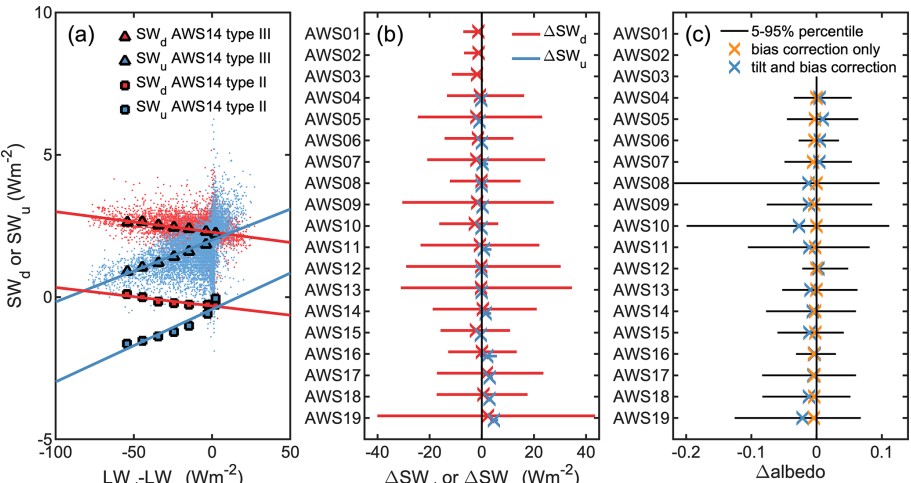

**Figure 5. (a)** Pyranometer zero offset: measured short-wave radiation components versus measured net long-wave radiation and corresponding bin averages and linear regression. Only nighttime data for solar zenith angles above 110° are shown. **(b)** Average (crosses) and 5–95th percentile range (lines) of the correction for both components of short-wave radiation during the entire measurement period per station. **(c)** Average (crosses) and 5–95th percentile range (lines) of the impact of the pyranometer bias and tilt correction on the hourly measured broadband short-wave albedo, only computed for data when the solar zenith angle is lower than 70°.

### 2.3.4   Pyranometer (short-wave radiation)

If not affected by rime of frost, the pyranometers used to measure short-wave radiation suffer from a zero offset due to net infrared cooling of the sensor and from tilt due to the imperfect levelling of the sensor boom. We correct for the zero offset using the following procedure. Per sensor we select nighttime periods with solar zenith angles larger than 110° and bin the recorded short-wave components as a function of

net long-wave radiation in windows of 5 W m$^{-2}$. We then fit an empirical linear regression to the averages in each bin, as described in Behrens (2021). An example of this procedure is shown in Fig. 5a. The correction is written as

$$\Delta SW_{u,d} = a_{u,d}LW_{net} + b_{u,d}, \tag{4}$$

with $\Delta SW_{u,d}$ the correction for either downward or upward short-wave radiation that is subtracted from the raw measurements, $LW_{net} = LW_d - LW_u$ the net recorded long-wave radi-

ation, and $a_{u,d}$ and $b_{u,d}$ empirically derived parameters of the linear regression that depend on the location and AWS type. This correction is then assumed to be valid continuously and is therefore applied to the entire dataset. On average, this correction does not exceed $5\,W\,m^{-2}$ (Fig. 5b).

To correct for the imperfect levelling of the AWS, we use the same empirical correction procedure for all stations, which is described by Van den Broeke et al. (2004). The method assumes that the measured daily totals of upward short-wave radiation are not affected by sensor inclination. An hourly "accumulated" albedo is then computed using the 24 h moving total $SW_u$ and $SW_d$ centred around the time of observation, which is then used to correct the hourly $SW_d$ from the hourly measured $SW_u$. This method was chosen for consistency and deemed most adequate given the lack of regular direct tilt measurements during AWS maintenance and varying tilt angles over time. However this method effectively removes the daily cycle in albedo. An alternative method was proposed by Wang et al. (2016) but was not applied to this dataset since this method requires knowledge of clear-sky days and a modelled surface radiation on these clear days.

The correction for both short-wave components ranges between $-5$ and $5\,W\,m^{-2}$ on average but varies between $-40$ and $40\,W\,m^{-2}$ for the downward component due to tilt (Fig. 5b). This correction also affects the estimated broadband short-wave albedo between $-0.02$ and $0.01$ on average depending on the station (Fig. 5c) but may reach between $-0.2$ and $0.1$ at AWS08 and AWS10, for example. Only a marginal fraction of the albedo correction is due to the zero offset.

### 2.3.5 Sonic height ranger

The readings from the sonic height ranger are corrected for the temperature dependence of the speed of sound according to

$$H_{corr} = H_{raw} \sqrt{\frac{T}{273.15}}, \tag{5}$$

with $H_{corr}$ and $H_{raw}$ the corrected and raw signals and $T$ the air temperature in Kelvin. The influence of humidity on the speed of sound is neglected due to the low humidity values and slower variation in air specific humidity.

### 2.4 Data processing/quality control

### 2.4.1 Filtering

Each measured variable is filtered using a threshold filter with bounds manually fixed to remove unrealistic values (Table A1). In addition, the measurements from the sonic height ranger are filtered with manually set thresholds that vary in time, after which the remaining individual spikes in the data are automatically removed using a moving median absolute difference filter. When no measurements are available from the data logger, due to logger failure, corrupted or missing data, or lack of maintenance visits, the data transmitted using Argos satellite communication are used instead, which are filtered for corrupted data generated by the satellite transmission. The data during periods with known logger/power supply issues or when the AWS structure was damaged by wind or buried by accumulation are also manually removed from the dataset. The dataset also contains a time series of filtered surface height after applying a daily moving average filter, which is used to estimate cumulative height change. The height of the sensor boom is manually reset during each maintenance visit and then estimated using the daily averaged filtered measurements from the sonic height ranger.

### 2.4.2 Derived variables

The list of variables contained in the dataset is given in Table 3. The specific humidity is computed as

$$q_v = \frac{RH}{100} q_s, \tag{6}$$

with RH the corrected measured relative humidity in percent and $q_s$ the equilibrium or saturated specific humidity, computed as

$$q_s = (R_d/R_v) \frac{e_s}{(p + e_s (R_d/R_v - 1))}, \tag{7}$$

with $R_v = 461.5\,J\,kg^{-1}\,K^{-1}$ and $R_d = 287.05\,J\,kg^{-1}\,K^{-1}$ the gas constants for water vapour and dry air, respectively; $p$ the air pressure; and $e_s$ the equilibrium water vapour pressure. The Magnus formula over ice is chosen for $e_s$, as given by WMO (2023):

$$e_s = 6.112 \exp\left[\frac{22.46T}{272.62 + T}\right], \tag{8}$$

with $T$ the air temperature in degrees centigrade, which is valid down to $-65\,°C$. This relationship is also used in this dataset for temperatures below this range but should be used with caution.

### 2.4.3 Correction to standard heights

The air temperature, specific humidity, and wind speed, corrected to standard heights above the surface, are also given in order to allow for an easy comparison between stations and with atmospheric models. Assuming the validity of Monin–Obukhov similarity theory, the quantities at standard heights

**Table 3.** Descriptions of variables in the dataset.

| Variable label | Unit or format | Description |
| --- | --- | --- |
| Time | yyyy-MM-dd HH:mm:ss | UTC time at the end of the measurement interval |
| $t$ | °C | Air temperature at boom height |
| t2m | °C | Air temperature corrected at 2 m height using similarity theory |
| $q$ | g kg$^{-1}$ | Specific humidity at boom height |
| q2m | g kg$^{-1}$ | Specific humidity corrected at 2 m height using similarity theory |
| rh | % | Relative humidity at boom height |
| rh2m | % | Relative humidity corrected at 2 m height using similarity theory |
| wspd | m s$^{-1}$ | Horizontal wind speed at boom height |
| wspd10m | m s$^{-1}$ | Horizontal wind speed corrected at 10 m height using similarity theory |
| wspdmax | m s$^{-1}$ | Maximum wind speed at boom height |
| wdir | ° | Wind from direction, positive clockwise with respect to true north |
| $p$ | hPa | Air pressure |
| SWd | W m$^{-2}$ | Downwards short-wave radiation |
| SWu | W m$^{-2}$ | Upwards short-wave radiation |
| LWd | W m$^{-2}$ | Downwards long-wave radiation |
| LWu | W m$^{-2}$ | Upwards long-wave radiation |
| z_surf | m | Unfiltered sonic height ranger measurement |
| z_surf_filtered | m | Filtered sonic height ranger measurement |
| cum_surface_height_zboom | m | Cumulative change in surface height since start from sonic height ranger on boom |
| LAT | decimal degrees | Latitude |
| LON | decimal degrees | Longitude |
| alb | – | Broadband short-wave albedo for solar zenith angles lower than 70° |
| Ts_obs | °C | Observed surface temperature assuming an emissivity of 1 |
| errorflag | rabcdefghi | Quality flag; see Sect. 2.4 and Table 4; 1 000 000 000 for non-suspicious data |

are written

$$u_{10\mathrm{m}} = \frac{u_*}{\kappa}\left[\ln\left(\frac{10}{z_{0\mathrm{m}}}\right) - \Psi_M\left(\frac{10}{L}\right) + \Psi_M\left(\frac{z_{0\mathrm{m}}}{L}\right)\right], \quad (9)$$

$$T_{2\mathrm{m}} = T_{\mathrm{s}} + \frac{\theta_*}{\kappa}\left[\ln\left(\frac{2}{z_{0T}}\right) - \Psi_H\left(\frac{2}{L}\right) + \Psi_H\left(\frac{z_{0T}}{L}\right)\right] - \frac{2g}{C_{\mathrm{p}}}, \quad (10)$$

$$q_{2\mathrm{m}} = q_{\mathrm{s}} + \frac{q_*}{\kappa}\left[\ln\left(\frac{2}{z_{0q}}\right) - \Psi_Q\left(\frac{2}{L}\right) + \Psi_Q\left(\frac{z_{0q}}{L}\right)\right], \quad (11)$$

where $z_{0\mathrm{m}}$ is the roughness length for momentum that is taken as a constant value of $10^{-4}$ m, and $z_{0T}$ and $z_{0q}$ are the scalar roughness lengths, which are parameterized after Andreas (1986). $T_{\mathrm{s}}$ is the recorded surface temperature, and $q_{\mathrm{s}}$ is the saturation specific humidity for $T = T_{\mathrm{s}}$. $\kappa = 0.4$ is the Von Kármán constant and $g = 9.81$ m s$^{-2}$ the gravitational acceleration. The Obukhov length $L$, as well as the fluxes $u_* = \sqrt{-\overline{u'w'}}$, $T_* = -\overline{w'T'}/u_*$, and $q_* = -\overline{w'q'}/u_*$, is computed using the bulk flux method implemented in the surface energy balance model described in Van Tiggelen et al. (2023), using the variable height of the sensors. The relations from Holtslag and De Bruin (1988) for the integrated stability functions $\Psi_M$, $\Psi_H$, and $\Psi_Q$ are used. The quantities at standard heights are only available when measurements of both the surface height and all four components of net radiation are not flagged.

### 2.4.4 Flagging

A binary quality flag is generated for each sample that aims to incorporate all the possible combinations of suspicious or missing data for each measured variable and possible riming or hoar frost deposition, hereafter denoted "riming", in one parameter (10 combinations). The flag is formatted as a combination of 10 consecutive 1's or 0's in the order of "rabcdefghi", where a numerical value of 1 denotes a suspicious or missing sample for a specific variable, and a numerical value of 0 denotes a seemingly properly functioning sensor. An exception is for the first value ("r"), which can have a value of 1 (no suspected riming) or 2 (suspected riming or cannot be excluded), in order to prevent an errorflag value with 10 consecutive 0's. The second-to-last binary flags are in respective order related to the following measurements: (a) the surface height, (b) air pressure, (c) specific humidity, (d) air temperature, (e) outgoing long-wave radiation, (f) incoming long-wave radiation, (g) outgoing short-wave radiation, (h) incoming short-wave radiation, and (i) wind speed. The specific conditions for which each flag is raised per variable are given in Table 4. In summary, valid samples when all the sensors seem to function properly at the same time have a value of 1 000 000 000. The resulting quality parameter allows for straightforward data selection or interpolation routines, which was not done in this dataset.

**Table 4.** Procedure for making the single quality parameter "errorflag" for flagging suspicious or missing data. TOA = top of atmosphere.

| Parameter in "errorflag" (rabcdefghi) | Description or flagged suspicious sample | Condition when flag raised |
|---|---|---|
| $r = 2$ | Riming suspected or cannot be excluded, or sensors are (close to) being buried | $|LW_{net}| < 2 \, W \, m^{-2}$ and RH > 90 % and $T < 0 \, °C$ for at least 24 consecutive hours, or any of these parameters are missing, or z_surf < 0.2 m |
| $a = 1$ | z_surf | z_surf missing or outside the manually fixed interval |
| $b = 1$ | $p$ | $p$ missing |
| $c = 1$ | $q$ | RH missing, or RH > 110 % |
| $d = 1$ | $t$ | $t$ missing |
| $e = 1$ | LWu | LWu missing, or LWu > 320 $W \, m^{-2}$, or $T < -60 \, °C$ |
| $f = 1$ | LWd | LWd missing, or LWu > 320 $W \, m^{-2}$, or $T < -60 \, °C$ |
| $g = 1$ | SWu | SWu missing, or daily total SWd exceeds TOA radiation, or daily total SWu is equal to zero when daily maximum TOA exceeds 10 $W \, m^{-2}$ |
| $h = 1$ | SWd | SWd missing, or daily total SWd exceeds TOA radiation, or daily total SWd is equal to zero when daily maximum TOA exceeds 10 $W \, m^{-2}$ |
| $i = 1$ | wspd | wspd missing, or wspd < 0.1 $m \, s^{-1}$ for at least 6 consecutive hours |

## 3 Data availability and range

The period covered by the dataset is shown in Fig. 6, and the number of valid, non-flagged samples per variable and per station is given in Table B1. In total, 56 757 d ($\approx$ 1.7 years) of data are available from 1995 through 2022, of which there were 36 000 d ($\approx$ 100 station years, or 65 %) of data with both unflagged meteorological observations, sonic height ranger measurements, and all four components of net surface radiation (Fig. 6). AWS09 has the longest time series (2.0 years), while AWS05 contains the longest time series of unflagged meteorological and SEB data (11.6 years). At AWS09, the outgoing long-wave radiation is missing between 2009 and 2016 due to a malfunctioning logger, while from 2017 onwards, the relative humidity is missing due to a malfunctioning sensor. This means that not more than 1 year of interpolated temperature data are available at AWS09 after 2009, since all variables need to be unflagged to reliably interpolate quantities using Eqs. (9)–(11). At the plateau sites AWS12 and AWS13, no pressure data is available due to a malfunctioning data logger, and no wind speed and wind direction data is available after 70 d of operation at AWS13 due to a malfunctioning sensor. Yet, despite the very cold and dry climate, most other variables including surface height are available for the entire period at AWS12 and AWS13. The type III AWSs have the highest success rate (95 %). Data gaps longer than several days often affect all the variables, due to either a malfunctioning logger or power supply (e.g. AWS14 during 2017) or the AWS being buried by snow (AWS11 in 2012). Long periods of long-wave radiation measurements are flagged at the plateau stations AWS09, AWS12, and AWS13 during winter, since the radiometers

function well below their operation temperature range, but are not removed from the dataset.

The time averages of all measurements of incoming radiation, air temperature, wind speed, and surface height change are plotted as a function of station elevation in Fig. 7, and the averages of all variables per station and per season are given in Table C1. On average, annual downward long-wave radiation decreases with elevation from 230 $W \, m^{-2}$ at the ice shelf locations to 100 $W \, m^{-2}$ at the plateau stations. The decrease in incoming long-wave radiation is compensated by a similar increase in incoming short-wave radiation from about 130 $W \, m^{-2}$ near sea level up to 200 $W \, m^{-2}$ on the Antarctic plateau. The range of yearly averaged air temperature contained in this dataset is $[-55; -15] \, °C$. The lowest daily average temperature at a sensor height of $-82.9 \, °C$ was recorded at AWS13 (Pole of inaccessibility) on 11 August 2010, and the highest daily averaged air temperature at sensor height of $10.2 \, °C$ was recorded at AWS18 (Cabinet Inlet, Larsen C ice shelf) on 26 May 2016. Temperatures above melting occur at 10 out of 19 stations, either on ice shelves (AWS04, 14, 15, 17, 18, 19) or close to ice shelves (AWS05), but also rarely at higher elevations on the grounded ice sheet (AWS06, 07, 16). The annual averaged wind speed at sensor height ranges between 3 and 8 $m \, s^{-1}$, with maximum daily averages ranging from 11.4 $m \, s^{-1}$ at AWS12 (Plateau Station B) up to 32.3 $m \, s^{-1}$ at AWS05 (Wasa/Aboa Camp Maudheimvida). The surface height change ranges from 92 $mm \, yr^{-1}$ of ice ablation at the Scharffenbergbotnen blue ice location AWS07 up to 2242 mm of snow accumulation per year at the Halvfarryggen ice rise location AWS11.

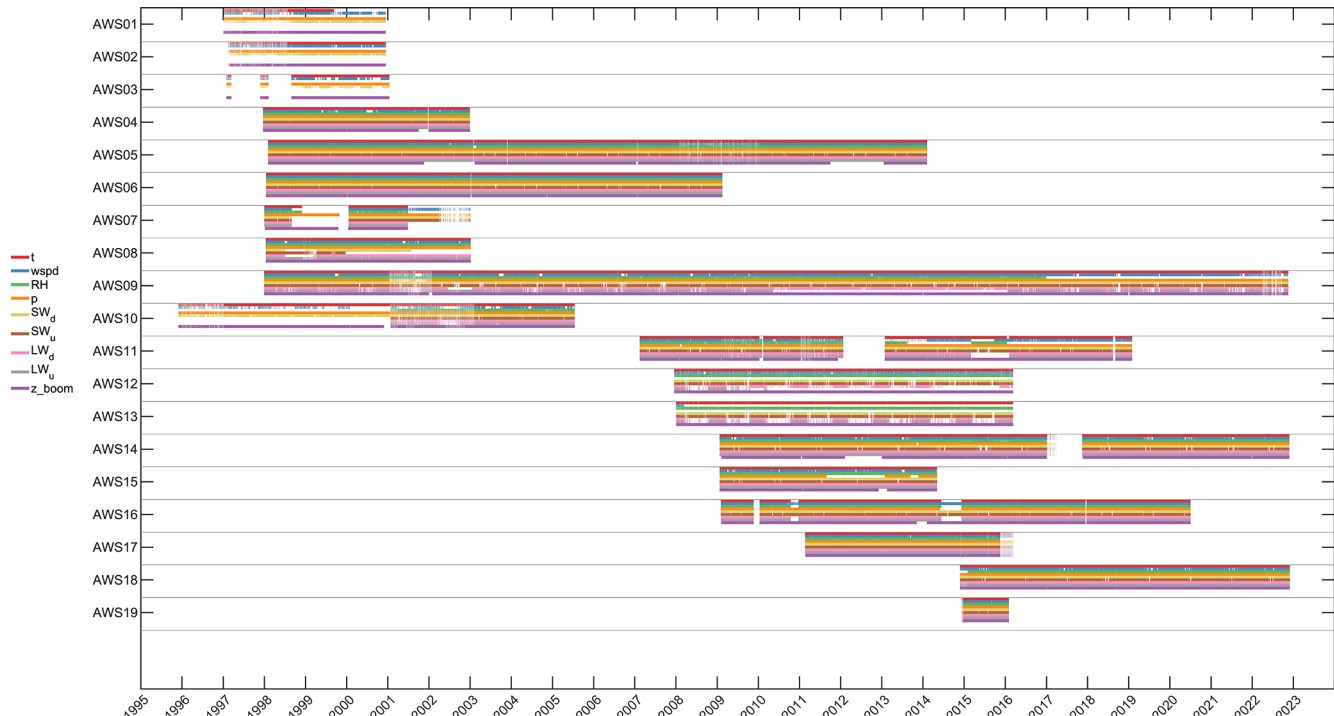

**Figure 6.** Availability of non-flagged hourly data per station. Each colour denotes a different variable. The number of non-flagged data per station and in total is also given. The number of non-flagged samples per variable and per station is given in Table B1.

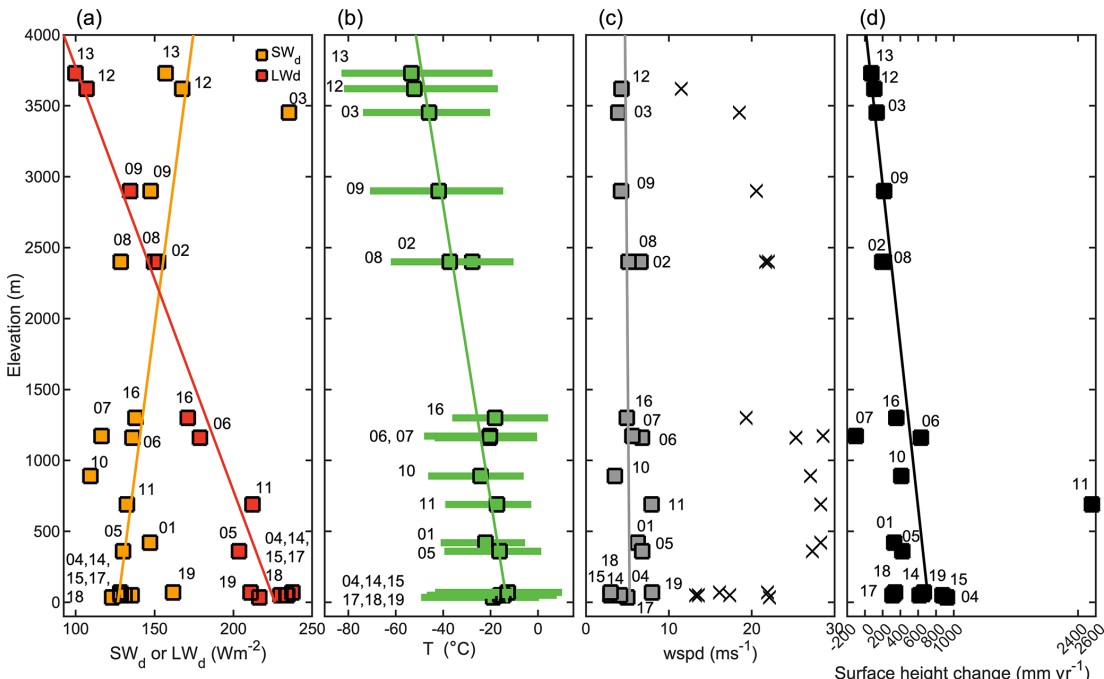

**Figure 7. (a)** Averaged downwards short-wave radiation (yellow), downward long-wave radiation (red), **(b)** average (squares) and minimum–maximum (lines) daily air temperature, **(c)** average (squares) and daily maximum (crosses) wind speed, **(d)** average increase in surface height for all the stations during the entire measurement period. Data are only shown when at least 50 % of data are available in the entire measurement period. Solid lines denote the linear regression versus elevation. The number of each AWS is also shown.

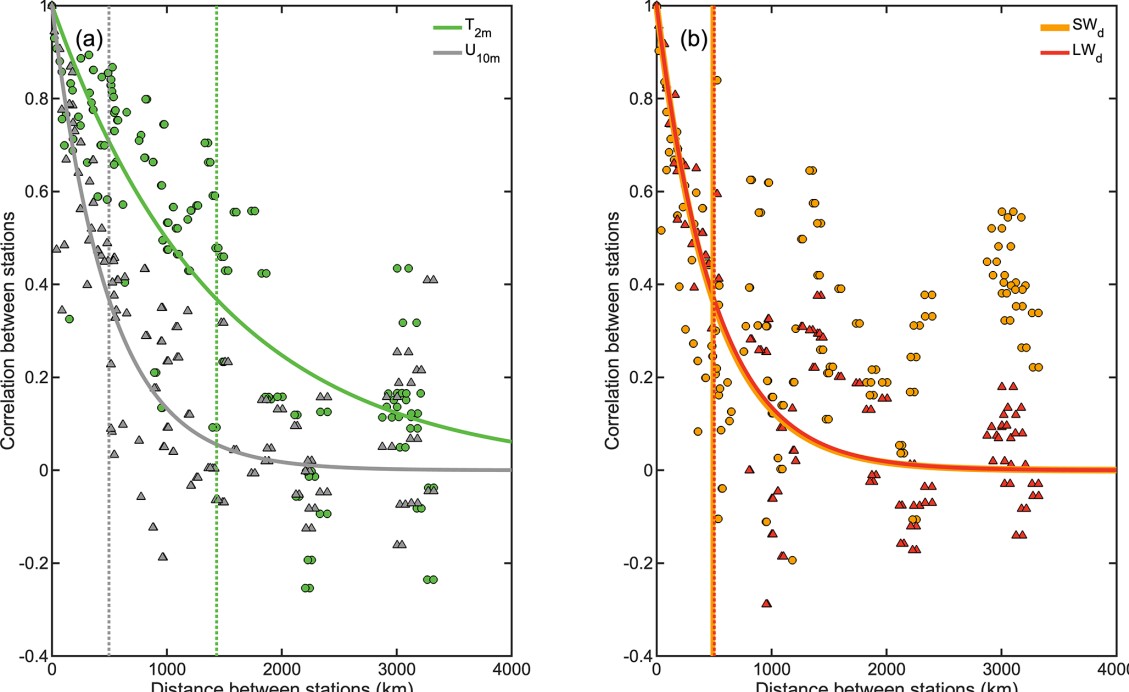

**Figure 8.** Temporal correlations of daily average **(a)** air temperature ($T$, green), horizontal wind speed ($U$, grey) and **(b)** downward short-wave radiation ($SW_d$, orange), and downward long-wave radiation ($LW_d$, red) in January between all station pairs versus the horizontal distance between stations. The lines denote the fit to exponential-decay functions, with the e-folding distance denoted by the vertical dashed lines.

## 4   Example application: inter-station correlation

An example application of this dataset is presented that takes advantage of the correlation of the measured variables between nearby stations, in order to spatially interpolate point measurements or design the optimal AWS network. The daily averaged measurements of temperature, wind speed, and both downward radiation components for January are selected. The correlation of each of these four variables is then computed for all station pairs during the periods of overlapping data and only retained if at least 1 month of overlapping data are available. The correlation for each station combination for each variable is then regressed as a function of the distance between all station pairs in Fig. 8. The distances and correlations are also given in Tables D1 and D2. The largest correlation is found for air temperature (Fig. 8a, green dots), with a fitted e-folding distance of 1430 km, which is defined as the distance where the fitted correlation equals 0.37. This distance is consistent with Bumbaco et al. (2014) and Hakim et al. (2020), who make use of different data. On the other hand, the correlation between stations for horizontal wind speed and for the two downward radiation components is lower (Fig. 8b), with an e-folding distance of about 500 km for all three variables. It must be noted that such correlations also depend on the region and on the season (Bumbaco et al., 2014). The lower correlation for wind speed and downward

radiation confirms the need for a denser array of stations for energy balance applications.

## 5   Data uncertainties and recommendations

While the corrections presented in Sect. 2.3 are based on short experiments with benchmark datasets, the weather stations have been in operation in remote areas on the Antarctic ice sheet, which prevents a direct comparison of their long-term measurements with independent benchmark datasets. As such, the uncertainty in the final corrected parameters is estimated as follows. From 1 January through 28 October 2001, a type I and a type II station were simultaneously operating at AWS10, and from 23 January through 31 December 2015, a type II and a type III station were simultaneously operational at AWS14. The mean absolute difference (MD) and centred root-mean-square deviation (RMSD) of the overlapping 2 h (AWS10) and hourly (AWS14) parameters are then computed and reported in Table 5. In addition, we compare the measured horizontal wind speed by the type II station at AWS14 with overlapping sonic anemometer data taken from 6 January through 30 January 2011 during the experiment described by Kuipers Munneke et al. (2012) and at AWS05 with overlapping sonic anemometer and fine-wire thermocouple data from 15 January 2007 through 5 July 2008. These additional data are not part of this dataset.

Finally, we compare both downwards radiation components measured by the type III station at AWS11 with independent reference data from the BSRN site at Neumayer from 14 February 2010 through 30 January 2019 (Schmithüsen, 2021), although these stations are located 80 km apart.

On average, the air temperature between stations differs by $0.76 \pm 0.91\,°C$ (MD $\pm$ RMSD) for stations I and II and $0.25 \pm 0.45°$ for stations II and III, which exceeds the reported accuracy by the manufacturer ($0.1\,°C$ for types I and III, $0.3\,°C$ for type II; Table 2). We attribute this larger uncertainty to the different sensor heights, to the different sampling strategy, and to the uncertainty in the excess temperature correction due to the variable effectiveness of the radiation shields. The comparison with a fine-wire thermocouple at AWS05 also reveals recorded air temperature differences of $0.2 \pm 0.64\,°C$, which confirms that the sensor's performance is slightly different than reported by the manufacturer.

The difference in measured, corrected relative humidity between station types II and III ($8\% \pm 14\%$) also exceeds the reported accuracy, which translates to specific humidity differences of $0.15\,g\,kg^{-1} \pm 0.32\,g\,kg^{-1}$, which remains limited due to the low temperatures in this dataset. The differences are mainly caused by the relative large uncertainty in the hygrometer readings at low temperatures (minimum of $-49.7\,°C$ at AWS14) and different sensor types, but also by different sensor heights and different radiation shield types.

The horizontal wind speed does not substantially differ from the reported manufacturer accuracy, except when comparing types I and II, which we attribute to the different sampling strategy and to the different response and overspeeding of cup anemometers. Further, the cup anemometer at AWS10 was never replaced after its installation in 1995. The comparison between type II and sonic anemometer data at AWS05 reveals differences of $0.1 \pm 0.85\,m\,s^{-1}$, which may be interpreted as the uncertainty in measured wind speed by stations II and III in the polar conditions from this dataset. We attribute the differences from sonic anemometer readings to different heights; flow distortion; poor response of propeller anemometers at low wind speeds (lower than $1.5\,m\,s^{-1}$); and noise in the sonic anemometer readings due to, for example, precipitation, blowing snow in the sensor volume, or riming of the transducer heads.

The mean absolute difference of all four radiative components ranges from 1 to $8\,W\,m^{-2}$ between stations. The performance of the Anderaa 2770 sensor used to measure downwards short-wave radiation on the type I stations is lower than reported by the manufacturer and also lower than the CNR1/4 used on types II and III, with a RMSD value of $34\,W\,m^{-2}$ versus $19\,W\,m^{-2}$. On the other hand, we find that the corrected hourly CNR1/4 readings differ by less than 5 % of the daily averages, which is better than reported by the manufacturer and consistent with the result from Van den Broeke et al. (2004). It must be noted that the location of AWS10 is more susceptible to frosting, which pos-

sibly causes larger downwards short-wave radiation differences compared to other locations, since different radiometers may behave differently under frosting conditions. Despite the 80 km horizontal separation and about 650 m elevation difference, downwards short-wave and long-wave radiation measured at AWS11 only differs from the benchmark measurements from the Neumayer BSRN site by $11 \pm 68$ and $4 \pm 31\,W\,m^{-2}$, respectively, on average over a 10-year period (Schmithüsen, 2021). This further supports the high accuracy of the corrected CNR1/4 readings. Small average differences in measured short-wave components may still translate into relatively large error in broadband short-wave albedo, with RMSD values of 0.026 at AWS14 during 2015.

Finally, the intercomparison of the hourly surface height recorded by two different sonic height rangers reveals RMSD values of 0.08 m between the type I/II stations and 0.04 m between type II/III stations. These values are higher than reported by the manufacturer, which we attribute to the horizontal variability in the snow surface and to noise generated by secondary reflections.

The measurements in this dataset were taken in (or in the vicinity of) Dronning Maud Land and the Larsen C ice shelf; hence different datasets should be included for Antarctica-wide studies. This may include other AWSs (e.g. Ding et al., 2022; Wang et al., 2023) or SMB (Favier et al., 2013; Wang et al., 2021) compilations; surface meteorological observations from year-round crewed stations; or surface radiation observations from the BSRN network at Neumayer (Schmithüsen, 2021), Concordia (Lupi et al., 2021), South Pole (Riihimaki et al., 2023), and Syowa (Ogawa et al., 2024).

For future Antarctic AWS design, we recommend including measurements of the four radiation components and surface height change, which allow for SEB quantification including melt and therefore for a more complete evaluation of climate models.

## 6   Data availability

The hourly dataset for all 19 AWSs is available at https://doi.org/10.1594/PANGAEA.974080 (Van Tiggelen et al., 2024).

## 7   Code availability

The code used to pre-process and correct measurement of temperature, long-wave radiation, and relative humidity for the specific instrument combinations used in this dataset is available at https://doi.org/10.5281/zenodo.15101447 (Van Tiggelen et al., 2025a). The code used to apply the other corrections, flag suspicious data, and compute new variables is available at https://doi.org/10.5281/zenodo.15058515 (Van Tiggelen et al., 2025b). The surface energy balance model used to compute the quantities at standard heights

**Table 5.** Estimated mean absolute difference (MD) and centred root-mean-square deviation (RMSD) of hourly measured parameters between two overlapping and independent datasets at five locations.

| Comparison between | type I and type II data | | type II and type III data | | type II and sonic anemometer data | | type II and Neumayer BSRN data | | type II and sonic anemometer with fine-wire thermocouple data | |
| | at AWS10 1 Jan 2001/ 28 Oct 2001 | | at AWS14 23 Jan 2015/ 31 Dec 2015 | | at AWS14 6 Jan 2011/ 30 Jan 2011 | | at AWS11 14 Feb 2010/ 30 Jan 2019 | | at AWS05 15 Jan 2007/ 5 Jul 2008 | |
| Variable | MD | RMSD | MD | RMSD | MD | RMSD | MD | RMSD | MD | RMSD |
|---|---|---|---|---|---|---|---|---|---|---|
| $t$ | 0.76 °C | 0.91 °C | 0.25 °C | 0.45 °C | | | | | 0.20 °C | 0.64 °C |
| rh | | | 8 % | 14 % | | | | | | |
| $q$ | | | 0.15 g kg$^{-1}$ | 0.32 g kg$^{-1}$ | | | | | | |
| wspd | 0.2 m s$^{-1}$ | 2.2 m s$^{-1}$ | 1.1 m s$^{-1}$ | 0.8 m s$^{-1}$ | 0.7 m s$^{-1}$ | 0.5 m s$^{-1}$ | | | 0.1 m s$^{-1}$ | 0.85 m s$^{-1}$ |
| $p$ | 132 Pa | 58 Pa | 26 Pa | 45 Pa | | | | | | |
| SWd | 2 W m$^{-2}$ | 34 W m$^{-2}$ | 8 W m$^{-2}$ | 19 W m$^{-2}$ | | | 11 W m$^{-2}$ | 68 W m$^{-2}$ | | |
| SWu | | | 5 W m$^{-2}$ | 14 W m$^{-2}$ | | | | | | |
| LWd | | | 5 W m$^{-2}$ | 6 W m$^{-2}$ | | | 4 W m$^{-2}$ | 31 W m$^{-2}$ | | |
| LWu | | | 1 W m$^{-2}$ | 2 W m$^{-2}$ | | | | | | |
| z_surf | 0.11 m | 0.08 m | 0.55 m | 0.04 m | 0.70 m* | | | | 0.22 m* | |
| alb | | | 0.002 | 0.026 | | | | | | |

* Same sonic height ranger data were used for the surface height change estimation of the sonic anemometer.

is available at https://doi.org/10.5281/zenodo.15082295 (Van Tiggelen et al., 2025c).

# 8 Summary

A dataset is presented that contains quality-controlled, continuous, hourly or 2 h measurements of meteorological quantities, net surface radiation components, and surface height change at 19 locations on the Antarctic ice sheet from 1995 through 2022. This dataset differs from other similar datasets containing these stations, such as Jakobs et al. (2020) and Wang et al. (2021), in how quality control and data corrections were implemented, but also in the number of available recorded parameters. The temperature, humidity, and radiation data are corrected for commonly documented measurements errors; noise was both manually and automatically removed; and missing data are left empty. In addition, the temperature, humidity, and wind at standard heights, as well as the cumulative change in surface height, are provided when available. A simple quality flag is provided for further use in, for example, surface energy balance simulations. Despite the remoteness and harsh climatic conditions of these locations, the average success rate of the hourly data is 90 % for the temperature, 74 % for the wind speed, 73 % for the four components of net surface radiation, 67 % for the filtered measured height change, and 50 % for all of the hourly data being simultaneously available and unflagged.

Some of the measurements were transmitted to the GTS system and are therefore available to be assimilated in reanalyses products. The exact periods and stations for which this is the case are marked in Table 1. The other measurements have not been assimilated and thereby allow for an indepen-

dent benchmark for reanalyses products. This dataset may also serve as basis for future work that requires in situ observations to study surface–climate interactions on the Antarctic ice sheet.

# Appendix A: Manual thresholds

**Table A1.** Manual thresholds applied to the dataset after the corrections. * These data were also processed with a manual threshold filter with time-varying bounds.

| Variable | Min | Max | Unit |
|---|---|---|---|
| $t$ | −90 | 30 | °C |
| rh | 0 | 150 | % |
| wspd | 0 | 100 | m s$^{-1}$ |
| wdir | 0 | 360 | ° |
| $p$ | 500 | 1100 | hPa |
| SWd | −20 | 1500 | W m$^{-2}$ |
| SWu | −20 | 1000 | W m$^{-2}$ |
| LWd | 50 | 500 | W m$^{-2}$ |
| LWu | 50 | 500 | W m$^{-2}$ |
| z_surf* | 0.1 | 30 | m |
| alb | 0.1 | 0.95 | – |

## Appendix B: Number of non-flagged data

**Table B1.** Number of days of valid, non-flagged data per variable and per station. All data denote the number of days with all variables being simultaneously non-flagged.

| Station | $t$ | wspd | RH | $p$ | SWd | SWu | LWd | LWu | z_boom | All data | Total days |
|---|---|---|---|---|---|---|---|---|---|---|---|
| AWS01 | 856 | 1089 | 0 | 1405 | 1149 | 0 | 0 | 0 | 1409 | 0 | 1409 |
| AWS02 | 1258 | 1183 | 0 | 1339 | 1157 | 0 | 0 | 0 | 1352 | 0 | 1352 |
| AWS03 | 963 | 713 | 0 | 987 | 598 | 0 | 0 | 0 | 991 | 0 | 991 |
| AWS04 | 1832 | 1718 | 1827 | 1832 | 1828 | 1828 | 1830 | 1830 | 1745 | 1626 | 1832 |
| AWS05 | 5810 | 5733 | 5787 | 5811 | 5700 | 5702 | 5809 | 5805 | 4885 | 4676 | 5821 |
| AWS06 | 4047 | 4045 | 4045 | 4047 | 3973 | 3973 | 4047 | 4047 | 4037 | 3960 | 4047 |
| AWS07 | 863 | 1108 | 830 | 1564 | 1131 | 1131 | 772 | 772 | 1180 | 720 | 1567 |
| AWS08 | 1819 | 1754 | 1814 | 1819 | 1060 | 657 | 1559 | 1782 | 1820 | 419 | 1820 |
| AWS09 | 9009 | 8568 | 6822 | 8872 | 8576 | 8577 | 7984 | 6242 | 9010 | 3683 | 9048 |
| AWS10 | 3307 | 1808 | 1496 | 3378 | 3207 | 1486 | 1463 | 1453 | 3383 | 979 | 3443 |
| AWS11 | 3899 | 3375 | 2246 | 3927 | 3767 | 3769 | 3582 | 3564 | 3866 | 1518 | 3955 |
| AWS12 | 2981 | 2707 | 2964 | 0 | 2683 | 2684 | 1605 | 429 | 3001 | 0 | 3001 |
| AWS13 | 2968 | 64 | 2943 | 0 | 2760 | 2763 | 1800 | 1798 | 2985 | 0 | 2987 |
| AWS14 | 4778 | 4589 | 4721 | 4758 | 4543 | 4540 | 4776 | 4776 | 4381 | 3966 | 4779 |
| AWS15 | 1931 | 1849 | 1917 | 1347 | 1885 | 1885 | 1930 | 1930 | 1855 | 1209 | 1931 |
| AWS16 | 3861 | 4098 | 3860 | 3914 | 4017 | 4016 | 3860 | 3861 | 4018 | 3669 | 4108 |
| AWS17 | 1798 | 1769 | 1725 | 1798 | 1798 | 1798 | 1796 | 1796 | 1793 | 1687 | 1798 |
| AWS18 | 2926 | 2816 | 2860 | 2926 | 2845 | 2845 | 2921 | 2921 | 2923 | 2667 | 2926 |
| AWS19 | 415 | 411 | 415 | 415 | 408 | 409 | 415 | 415 | 415 | 404 | 415 |
| Total | 55 321 | 49 397 | 46 272 | 50 139 | 53 085 | 48 063 | 46 149 | 43 421 | 55 049 | 31 183 | 57 230 |

## Appendix C: Station climatology

**Table C1.** Long-term (AVG), summer (DJM), and winter (JJA) averages of the hourly variables in this dataset: air temperature ($t$), specific humidity (qv), relative humidity (rh), horizontal wind speed (wspd), air pressure ($p$), downwards short-wave radiation (LWd), upward long-wave radiation (LWu), broadband short-wave albedo for solar zenith angles lower than 70° (alb), and surface temperature assuming unit emissivity (Ts). The percentage of valid hourly data is also given for each variable.

| Variable Unit | | $t$ °C | qv g kg$^{-1}$ | rh % | wspd m s$^{-1}$ | $p$ hPa | SWd W m$^{-2}$ | SWu W m$^{-2}$ | LWd W m$^{-2}$ | LWu W m$^{-2}$ | alb – | Ts °C |
|---|---|---|---|---|---|---|---|---|---|---|---|---|
| AWS01 | AVG | −22.3 | – | – | 6.2 | 817 | 149 | 0 | – | – | – | – |
| | JJA | −27.2 | – | – | 7.7 | 815 | 7 | 0 | – | – | – | – |
| | DJF | −13.2 | – | – | 4.8 | 822 | 339 | 0 | – | – | – | – |
| | % data | 61 | 0 | 0 | 77 | 100 | 85 | 0 | 0 | 0 | 0 | 0 |
| AWS02 | AVG | −27.8 | – | – | 6.6 | 713 | 151 | 0 | – | – | – | – |
| | JJA | −31.8 | – | – | 7.8 | 710 | 5 | 0 | – | – | – | – |
| | DJF | −19.9 | – | – | 5.0 | 720 | 379 | 0 | – | – | – | – |
| | % data | 94 | 0 | 0 | 88 | 99 | 92 | 0 | 0 | 0 | 0 | 0 |
| AWS03 | AVG | −46.4 | – | – | 4.1 | 617 | 231 | 0 | – | – | – | – |
| | JJA | −56.4 | – | – | 3.8 | 615 | 10 | 0 | – | – | – | – |
| | DJF | −33.4 | – | – | 4.3 | 624 | 416 | 0 | – | – | – | – |
| | % data | 97 | 0 | 0 | 72 | 100 | 95 | 0 | 0 | 0 | 0 | 0 |
| AWS04 | AVG | −18.9 | 1.04 | 94.6 | 5 | 979 | 123 | 107 | 216 | 236 | 0.88 | −18 |
| | JJA | −26.7 | 0.48 | 96.1 | 4.8 | 980 | 3 | 3 | 199 | 208 | – | −24.9 |
| | DJF | −7 | 2.09 | 91.9 | 4.9 | 980 | 285 | 247 | 246 | 281 | 0.87 | −8.2 |
| | % data | 100 | 100 | 100 | 100 | 100 | 100 | 100 | 100 | 100 | 18 | 75 |
| AWS05 | AVG | −16.1 | 1.04 | 83.0 | 6.8 | 943 | 130 | 108 | 204 | 240 | 0.84 | −18.6 |
| | JJA | −22.9 | 0.52 | 84.3 | 7.3 | 943 | 3 | 3 | 184 | 213 | – | −25.7 |
| | DJF | −6.9 | 1.90 | 80.9 | 5.4 | 946 | 308 | 254 | 229 | 278 | 0.83 | −8.6 |
| | % data | 100 | 99 | 99 | 99 | 100 | 100 | 100 | 100 | 100 | 18 | 80 |
| AWS06 | AVG | −20.4 | 0.75 | 78.6 | 6.8 | 855 | 137 | 114 | 179 | 224 | 0.84 | −23.9 |
| | JJA | −26.6 | 0.39 | 80.1 | 7.6 | 853 | 2 | 2 | 165 | 200 | – | −29.6 |
| | DJF | −11.5 | 1.35 | 77.0 | 5.5 | 860 | 322 | 266 | 201 | 258 | 0.83 | −13.6 |
| | % data | 100 | 100 | 100 | 100 | 100 | 100 | 100 | 100 | 100 | 17 | 97 |
| AWS07 | AVG | −20.3 | 0.51 | 61.9 | 5.7 | 847 | 111 | 69 | 182 | 228 | 0.62 | −21.9 |
| | JJA | −26.4 | 0.29 | 68.9 | 6.2 | 846 | 4 | 3 | 168 | 205 | – | −28.4 |
| | DJF | −9.9 | 0.90 | 47.7 | 4.8 | 852 | 274 | 168 | 203 | 270 | 0.62 | −11.0 |
| | % data | 55 | 54 | 54 | 71 | 100 | 73 | 73 | 49 | 49 | 12 | 6 |
| AWS08 | AVG | −37.2 | 0.26 | 94.8 | 5.1 | 719 | 128 | 113 | 149 | 175 | 0.82 | −37.9 |
| | JJA | −45.1 | 0.1 | 97.5 | 5.5 | 717 | 1 | 1 | 150 | 153 | – | −41.1 |
| | DJF | −24.4 | 0.60 | 89.7 | 4.4 | 723 | 334 | 294 | 159 | 215 | 0.81 | −28.0 |
| | % data | 100 | 100 | 100 | 100 | 100 | 59 | 37 | 86 | 98 | 4 | 16 |
| AWS09 | AVG | −41.8 | 0.19 | 91.6 | 4.2 | 677 | 147 | 120 | 135 | 161 | 0.82 | −39.5 |
| | JJA | −51.9 | 0.06 | 92.1 | 4.4 | 673 | 2 | 2 | 135 | 137 | – | −48.3 |
| | DJF | −27.5 | 0.46 | 88.6 | 4.2 | 682 | 354 | 288 | 142 | 199 | 0.82 | −30.4 |
| | % data | 100 | 72 | 74 | 99 | 98 | 100 | 100 | 95 | 73 | 17 | 31 |
| AWS10 | AVG | −24.3 | 0.65 | 96.9 | 3.4 | 880 | 110 | 98 | 200 | 216 | 0.89 | −21.5 |
| | JJA | −31.2 | 0.28 | 96.0 | 2.2 | 878 | 3 | 1 | 187 | 190 | – | −33.5 |
| | DJF | −14.2 | 1.36 | 94.6 | 4.0 | 884 | 279 | 257 | 215 | 253 | 0.89 | −14.8 |
| | % data | 96 | 40 | 40 | 61 | 98 | 95 | 41 | 39 | 39 | 6 | 21 |

| Variable<br>Unit | | $t$<br>°C | $qv$<br>g kg$^{-1}$ | rh<br>% | wspd<br>m s$^{-1}$ | $p$<br>hPa | SWd<br>W m$^{-2}$ | SWu<br>W m$^{-2}$ | LWd<br>W m$^{-2}$ | LWu<br>W m$^{-2}$ | alb<br>– | Ts<br>°C |
|---|---|---|---|---|---|---|---|---|---|---|---|---|
| AWS11 | AVG | −17.4 | 0.94 | 89.7 | 8.0 | 902 | 133 | 115 | 212 | 237 | 0.87 | −20.0 |
| | JJA | −23.4 | 0.52 | 91.3 | 7.7 | 901 | 6 | 5 | 195 | 212 | – | −26.7 |
| | DJF | −9.2 | 1.75 | 87.0 | 7.1 | 904 | 306 | 264 | 235 | 270 | 0.87 | −11.5 |
| | % data | 99 | 56 | 57 | 85 | 99 | 100 | 100 | 91 | 90 | 19 | 33 |
| AWS12 | AVG | −52.2 | 0.11 | 91.8 | 4.3 | – | 168 | 131 | 107 | 131 | 0.78 | – |
| | JJA | −63.7 | 0.02 | 90.2 | 4.4 | – | 1 | 1 | 109 | 112 | – | – |
| | DJF | −35.1 | 0.32 | 92.4 | 4 | – | 401 | 311 | 111 | 169 | 0.78 | – |
| | % data | 99 | 99 | 99 | 92 | 0 | 99 | 99 | 82 | 23 | 16 | 0 |
| AWS13 | AVG | −53.4 | 0.09 | 89.0 | 4.4 | – | 157 | 121 | 100 | 130 | 0.76 | – |
| | JJA | −64.3 | 0.01 | 88.0 | – | – | 0 | 0 | 106 | 108 | – | – |
| | DJF | −36.3 | 0.27 | 87.9 | 4.2 | – | 393 | 298 | 100 | 170 | 0.76 | – |
| | % data | 99 | 99 | 99 | 2 | 0 | 99 | 99 | 98 | 98 | 15 | 0 |
| AWS14 | AVG | −15 | 1.34 | 93.1 | 3.4 | 985 | 131 | 110 | 233 | 250 | 0.85 | −14.3 |
| | JJA | −24 | 0.58 | 95.5 | 3 | 988 | 10 | 9 | 208 | 216 | – | −23.4 |
| | DJF | −4.2 | 2.51 | 89.8 | 3.5 | 983 | 276 | 232 | 268 | 294 | 0.84 | −4.8 |
| | % data | 100 | 98 | 99 | 100 | 100 | 100 | 100 | 100 | 100 | 20 | 77 |
| AWS15 | AVG | −16 | 1.28 | 94.9 | 3 | 986 | 128 | 114 | 230 | 246 | 0.89 | −15.9 |
| | JJA | −24.6 | 0.54 | 96.1 | 2.6 | 989 | 9 | 8 | 206 | 214 | – | −24.8 |
| | DJF | −4.9 | 2.47 | 93.2 | 3.1 | 985 | 273 | 239 | 265 | 289 | 0.88 | −6.0 |
| | % data | 100 | 99 | 99 | 100 | 70 | 100 | 100 | 100 | 100 | 20 | 56 |
| AWS16 | AVG | −18.1 | 0.69 | 61 | 4.9 | 827 | 189 | 114 | 171 | 218 | 0.82 | −24.7 |
| | JJA | −23.6 | 0.38 | 62.2 | 5.6 | 824 | 5 | 4 | 154 | 195 | – | −31.3 |
| | DJF | −10.3 | 1.28 | 65.5 | 4.4 | 832 | 323 | 263 | 200 | 256 | 0.82 | −14.2 |
| | % data | 94 | 94 | 94 | 100 | 95 | 100 | 100 | 94 | 94 | 17 | 89 |
| AWS17 | AVG | −14.9 | 1.25 | 91.8 | 4.1 | 987 | 133 | 109 | 233 | 251 | 0.83 | −16.0 |
| | JJA | −23.6 | 0.56 | 94.6 | 4 | 988 | 14 | 11 | 208 | 217 | 0.81 | −24.5 |
| | DJF | −4.4 | 2.44 | 89.8 | 3.7 | 984 | 278 | 230 | 268 | 295 | 0.83 | −4.8 |
| | % data | 100 | 96 | 96 | 100 | 100 | 100 | 100 | 100 | 100 | 22 | 92 |
| AWS18 | AVG | −12.7 | 1.40 | 84.9 | 3 | 981 | 128 | 110 | 237 | 256 | 0.86 | −13.2 |
| | JJA | −20.9 | 0.74 | 89.1 | 2.7 | 984 | 11 | 10 | 213 | 223 | – | −21.8 |
| | DJF | −3.7 | 2.35 | 81.0 | 2.7 | 980 | 272 | 229 | 268 | 296 | 0.85 | −4.6 |
| | % data | 100 | 98 | 98 | 100 | 100 | 100 | 100 | 100 | 100 | 20 | 85 |
| AWS19 | AVG | −13 | 1.19 | 80.8 | 8 | 978 | 160 | 129 | 211 | 257 | 0.81 | −14.1 |
| | JJA | −21.4 | 0.64 | 95.0 | 10 | 974 | 5 | 4 | 188 | 223 | – | −22.9 |
| | DJF | −4.4 | 1.83 | 67.4 | 4.4 | 980 | 321 | 251 | 238 | 292 | 0.79 | −5.3 |
| | % data | 100 | 100 | 100 | 100 | 100 | 100 | 100 | 100 | 100 | 22 | 97 |

## Appendix D: Station distances and correlation

**Table D1.** Horizontal distance between each station in kilometres.

| AWS | 02 | 03 | 04 | 05 | 06 | 07 | 08 | 09 | 10 | 11 | 12 | 13 | 14 | 15 | 16 | 17 | 18 | 19 |
|-----|----|----|----|----|----|----|----|----|----|----|----|----|----|----|----|----|----|----|
| 01 | 40 | 513 | 634 | 561 | 552 | 544 | 573 | 361 | 1532 | 358 | 1172 | 1640 | 2423 | 2406 | 698 | 2509 | 2525 | 824 |
| 02 | | 489 | 617 | 541 | 523 | 514 | 536 | 321 | 1495 | 359 | 1144 | 1607 | 2403 | 2385 | 698 | 2490 | 2505 | 828 |
| 03 | | | 962 | 878 | 774 | 758 | 650 | 432 | 1498 | 816 | 661 | 1153 | 2640 | 2612 | 429 | 2737 | 2738 | 576 |
| 04 | | | | 85 | 229 | 247 | 427 | 541 | 1094 | 347 | 1497 | 1836 | 1802 | 1785 | 1290 | 1889 | 1905 | 1428 |
| 05 | | | | | 162 | 178 | 359 | 456 | 1103 | 305 | 1416 | 1765 | 1873 | 1854 | 1208 | 1961 | 1976 | 1346 |
| 06 | | | | | | 18 | 198 | 343 | 1011 | 398 | 1270 | 1604 | 1914 | 1890 | 1135 | 2008 | 2016 | 1279 |
| 07 | | | | | | | 180 | 327 | 1011 | 403 | 1251 | 1586 | 1927 | 1903 | 1121 | 2022 | 2029 | 1265 |
| 08 | | | | | | | | 251 | 956 | 537 | 1083 | 1405 | 2006 | 1978 | 1044 | 2105 | 2106 | 1192 |
| 09 | | | | | | | | | 1197 | 478 | 977 | 1377 | 2236 | 2210 | 805 | 2331 | 2336 | 952 |
| 10 | | | | | | | | | | 1381 | 1578 | 1582 | 1450 | 1398 | 1902 | 1566 | 1532 | 2045 |
| 11 | | | | | | | | | | | 1447 | 1865 | 2123 | 2110 | 1056 | 2204 | 2226 | 1181 |
| 12 | | | | | | | | | | | | 526 | 2913 | 2874 | 814 | 3018 | 3003 | 889 |
| 13 | | | | | | | | | | | | | 2973 | 2927 | 1330 | 3083 | 3054 | 1398 |
| 14 | | | | | | | | | | | | | | 67 | 3075 | 122 | 108 | 3215 |
| 15 | | | | | | | | | | | | | | | 3045 | 183 | 141 | 3186 |
| 16 | | | | | | | | | | | | | | | | 3120 | 3126 | 151 |
| 17 | | | | | | | | | | | | | | | | | 86 | 3320 |
| 18 | | | | | | | | | | | | | | | | | | 3323 |
| 19 | | | | | | | | | | | | | | | | | | |

**Table D2.** Temporal correlation of hourly January air temperature between each station with at least 1 month of overlapping data.

| AWS | 02 | 03 | 04 | 05 | 06 | 07 | 08 | 09 | 10 | 11 | 12 | 13 | 14 | 15 | 16 | 17 | 18 | 19 |
|-----|----|----|----|----|----|----|----|----|----|----|----|----|----|----|----|----|----|----|
| 01 | 0.91 | 0.83 | 0.40 | 0.66 | 0.77 | 0.73 | 0.75 | 0.86 | 0.43 | – | – | – | – | – | – | – | – | – |
| 02 | | 0.86 | 0.57 | 0.77 | 0.81 | 0.84 | 0.8 | 0.89 | 0.46 | – | – | – | – | – | – | – | – | – |
| 03 | | | 0.5 | 0.66 | 0.72 | 0.71 | 0.77 | 0.85 | 0.23 | – | – | – | – | – | – | – | – | – |
| 04 | | | | 0.86 | 0.76 | 0.74 | 0.70 | 0.66 | 0.52 | – | – | – | – | – | – | – | – | – |
| 05 | | | | | 0.83 | 0.82 | 0.78 | 0.70 | 0.47 | 0.66 | 0.59 | 0.56 | 0.15 | 0.16 | 0.56 | 0.16 | – | – |
| 06 | | | | | | 0.93 | 0.86 | 0.79 | 0.53 | 0.59 | 0.57 | 0.56 | – | – | – | – | – | – |
| 07 | | | | | | | 0.69 | 0.81 | 0.48 | – | – | – | – | – | – | – | – | – |
| 08 | | | | | | | | 0.89 | 0.61 | – | – | – | – | – | – | – | – | – |
| 09 | | | | | | | | | 0.43 | 0.58 | 0.74 | 0.66 | −0.01 | 0.00 | 0.67 | −0.09 | 0.13 | 0.13 |
| 10 | | | | | | | | | | – | – | – | – | – | – | – | – | – |
| 11 | | | | | | | | | | | 0.48 | 0.42 | −0.06 | 0.12 | 0.57 | −0.25 | −0.19 | 0.54 |
| 12 | | | | | | | | | | | | 0.87 | 0.14 | 0.11 | 0.80 | 0.11 | 0.43 | 0.21 |
| 13 | | | | | | | | | | | | | 0.15 | 0.16 | 0.7 | 0.17 | 0.32 | 0.09 |
| 14 | | | | | | | | | | | | | | 0.88 | 0.05 | 0.77 | 0.70 | −0.08 |
| 15 | | | | | | | | | | | | | | | 0.17 | 0.71 | – | – |
| 16 | | | | | | | | | | | | | | | | 0.09 | 0.12 | 0.33 |
| 17 | | | | | | | | | | | | | | | | | 0.76 | −0.24 |
| 18 | | | | | | | | | | | | | | | | | | −0.04 |
| 19 | | | | | | | | | | | | | | | | | | |

**Author contributions.** Data curation: PS, CHR, and MVT. Data corrections: PS and MVT. Formal analysis: MVT with the help of all the authors. Writing/visualization: MVT with the help of all the authors. Funding: CHR and MRVB with the help of all the authors. Conceptualization/investigation/methodology: all the authors. Project administration: CHR and MRVB.

**Competing interests.** The contact author has declared that none of the authors has any competing interests.

**Disclaimer.** Publisher's note: Copernicus Publications remains neutral with regard to jurisdictional claims made in the text, published maps, institutional affiliations, or any other geographical representation in this paper. While Copernicus Publications makes ev-

ery effort to include appropriate place names, the final responsibility lies with the authors.

**Acknowledgements.** The authors thank all the persons involved in the transport, installation, servicing, and dismantling of the weather stations. The technical staff of the IMAU is acknowledged for the design of the weather stations. The Alfred Wegener Institute (AWI) is acknowledged for their support regarding the weather stations at Kohnen Base (AWS09), on Berkner Island (AWS10), and on the Halvfarryggen ice rise (AWS11). The installations of weather stations on Plateau Station B (AWS12) and the Pole of Inaccessibility (AWS13) were carried out under the umbrella of the project Trans-Antarctic Scientific Traverse Expeditions – Ice Divide of East Antarctica (TASTE-IDEA), funded by the Norwegian Polar Institute, the US National Science Foundation (NSF), and the Research Council of Norway within the framework of TASTE-IDEA project 152 of IPY 2007–2008. The installations of AWS01-03 were carried out at a traverse during the Norwegian Antarctic Research Expedition (NARE) 1996/1997 organized by the Norwegian Polar Institute. The traverse was a contribution to the "European Project for Ice Coring in Antarctica" (EPICA), a joint ESF (European Science Foundation)/EC scientific programme, funded by the European Commission under the Environment and Climate Programme (1994–01998) contract ENV4-CT95-0074 and by national contributions from Belgium, Denmark, France, Germany, Italy, the Netherlands, Norway, Sweden, Switzerland, and the United Kingdom. The Finnish Antarctic Research Program (FINNARP) at the Finnish Meteorological Institute (FMI) is acknowledged for their support in the operation of the weather station at Wasa/Aboa Camp Maudheimvida (AWS5). The Royal Meteorological Institute of Belgium (KMI) and KU Leuven are acknowledged for their support regarding the weather station at the Princess Elisabeth station (AWS16) and on the King Baudouin ice shelf (AWS19). The British Antarctic Survey is acknowledged for their support with the operations of the weather stations on the Larsen C ice shelf – North (AWS14), South (AWS15), and Cabinet inlet (AWS18) – and on the remnants of the Larsen B ice shelf on Scar Inlet (AWS17). The Swedish Antarctic Research Programme (SWEDARP) is acknowledged for their support with the operations of the weather station at the Rampen site 1090 (AWS04), Wasa/Aboa Camp Maudheimvida (AWS05), Svea (AWS06), Scharffenbergbotnen (AWS07), and Camp Victoria (AWS08) in Dronning Maud Land. The weather station on Roi Baudouin ice shelf (AWS19) was installed within the framework of the BENEMELT project, funded by the InBev-Baillet Latour Antarctica Fellowship, a joint initiative of the InBev-Baillet Latour Fund and the International Polar Foundation (IPF). The authors thank PANGAEA for hosting the dataset.

**Financial support.** This work is funded by the Dutch Research Council (NWO) projects "Dutch Polar Climate and Cryosphere Change Consortium" (DP4C, no. ALWPP.2019.003) and "State and fate of Antarctica's gatekeepers: a HIgh Resolution approach for Ice ShElf instability" (HiRISE, no. OCENW.GROOT.2019.091).

**Review statement.** This paper was edited by Charles Amory and reviewed by Ian Allison, David Bromwich, and one anonymous referee.

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
