# Peer review of "IMAU Antarctic automatic weather station data, including surface radiation balance (1995-2022)"

_Earth System Science Data, 2025_

## Referee Comment (RC2)

**Comments on "IMAU Antarctic automatic weather station data,**

**including surface radiation balance (1995-2022)"**

Given the critical impact of the Antarctic ice sheet on global climate change and sea level rise, it is urgent to develop an improved in situ meteorological observation network. This manuscript reported almost 30 years of Antarctic surface meteorological observations from 19 Automated Weather Stations (AWS) operated by the Institute for Marine and Atmospheric research Utrecht (IMAU) at Utrecht University, including especially the surface energy and mass balance. The manuscript described in detail the variables, instrumentation, and processing of the observational data, and ultimately produced an accessible dataset of great importance to Antarctic climate change and surface mass balance studies. How the data were preprocessed and corrected were also described with detail in the manuscript. This is important for the assessment and development of regional climate models for the Antarctic Ice Sheet, the validation of remote sensing as well as reanalysis products, and contributes to an increased understanding of Antarctic surface climatology. The subject of this work is aligned with Earth System Science Data and is very valuable. I recommend publishing it in ESSD with minor revisions.

 The introduction is written concisely and logically, but it is short enough to provide more background to the study. I would suggest that the authors give an appropriate description about the important role of Antarctica in the global climate system, as well as adding a more detailed overview of the meteorological observation networks that are currently being set up. This could include more detail on some of the projects and programs already mentioned.

- 2. Regarding the radiative component, some of the short-wave radiation in the previously released data (https://doi.pangaea.de/10.1594/PANGAEA.910473) was observed during the winter months (most likely during the polar night), and I am not sure whether this is due to instrumental error or some other phenomenon (may not be an error, but an objective phenomenon). I would like to know if this problem still exists in the range reported in this manuscript.
- 3. -Another similar problem is that some of the relative humidity in the previous data exceeded 100%. How this data was processed in this work?
- 4. There are some acronyms that appear multiple times in the manuscripts, such as the British Antarctic Survey (BAS), and whether it is possible to retain them only in their first appearance.
- L66-67: Is it possible to correct for the effects of this positional movement on meteorological observations? This would be much more useful for the use of the data.
- Figure 1: I think it would be useful to add a wind vector scale to measure wind speed.

-Similarly, it is suggested that climatological values for other atmospheric variables be given, which can add the reader's understanding of the Antarctic surface climatology. They can be placed in a supplementary file.

---

## Referee Comment (RC4)

**General Comments**

This preprint provides an excellent description of more than 20 years of high quality and valuable Antarctic surface meteorological data from the IMAU automatic weather station (AWS) network. These AWS were designed to enable estimation of the ice sheet surface energy and mass balances.The publication clearly outlines what variables were measured, what instrumentation was used, how the data were processed and how corrections were made. The accuracy of the measurements is assessed in several ways including during periods when overlapping measurements from two AWS were made at five of the locations. The data availability from each station is summarised in both Figure and Table and the climatology at each site is also summarized.

Clear links are given to how the data can be accessed and to the software codes used to pre-process and correct the measurements. I particularly liked the simple flag assigned to each data sample alerting of potential problems in each of the measured variables.

The IMAU Antarctic AWS network is one of several that provide Antarctic surface meteorological data. Others include those of the University of Wisconsin, the Australian Antarctic Division and the Chinese Antarctic Programme. These are mentioned and acknowledged in the preprint. But I think that the larger Antarctic AWS data set can be more directly referenced with a few small changes that do not length the manuscript (the focus of which clearly should be the IMAU network). This could be simply done by noting in the Introduction that the locations of all AWS that have been deployed in Antarctica are shown on Figure 1 and that details of these stations are given in Wang et al., 2023 (reference already cited).

I could find no mention of what height the sensors were initially deployed at: this should be included around line 76. Also, while it is implicit that the wind, temperature and humidity data are corrected for height changes from snow accumulation/ablation, that is not explicitly noted until line 240. Perhaps a comment should be added in the introduction to the effect that "after correction for sensor errors and surface height change from snow accumulation/ablation, the wind, temperature and humidity data are corrected to standard heights for SEB calculations using Monin-Obukhovsimilarity theory."

This manuscript clearly fits the objectives and standards of ESSD.I think it should be published after only minor changes and consideration of the following specific and technical comments.

**Specific Comments (referenced to line number)**

| 68      | A brief description here of what is done during maintenance visits would be useful.
Some details are given later in 2.4. A problem we experienced with AWS high on the          |
|---------|------------------------------------------------------------------------------------------------------------------------------------------------------------------------------------|
|         | plateau is that the acoustic lens of the Cambel SR50 failed after a year or so in very cold conditions. Was that a problem with IMAU AWS and, if so, was it fixed during           |
|         | maintenance?                                                                                                                                                                       |
| Table 2 | I think that this table should be relocated after Section 2.2. Reading Table 2 first I was                                                                                         |
|         | left wondering what sort of radiation shields were used, whether the anemometer model numbers refer to propellor or vane anemometers, etc. These are addressed in the              |
|         | text.                                                                                                                                                                              |
| Table 2 | What depth is snow temperature measured at? Is it also corrected for subsequent accumulation?                                                                                      |
| 100     | Riming or hoar frost deposition? Inland station sensors are usually affected by hoar frost deposition (ice crystals formed by direct sublimation of water vapour) rather than rime |

(ice formed from freezing of supercooled liquid water). Coastal stations may suffer riming.

- 118 Does "hut temperature" mean "temperature inside the radiation shield"? I think the use of "hut" is inappropriate.
- 153 & 186 Largest radiation measurement errors probably occur with hoar frost deposition on the instruments. These events are flagged, but no corrections are made. An additional note regarding this might be added here.
- 314 & 332 Were thesonic anemometersused at the AWS14 and AWS05 site comparisonsseparate instruments, not part of any AWS?

**Technicaland minor comments (referenced to line number)**

| 10           | This seems unnecessary as a reference as it just refers to the url given in the preceding              |
|--------------|--------------------------------------------------------------------------------------------------------|
|              | line (https://doi.pangaea.de/10.1594)                                                                  |
| 51           | "International Polar Year (IPY 2007-2008)"                                                             |
| 54           | "one AWS was installed at former"                                                                      |
| Fig 1        | Location labelling of AWS 04, 05, 06, 07, 08, 09 is not clear in the main picture, but is              |
|              | clarified in the inset                                                                                 |
| 58           | "CIRES and the the let Propulsion Laboratory"                                                          |
| 59           | "in SCAR Inlet"                                                                                        |
| 62           | "the Belgian Royal Meteorological Institute" or "the Royal Meteorological Institute of                 |
|              | Belgium"                                                                                               |
| 63           | "Belgian Princess Elizabeth station"                                                                   |
| Table 1      | "and renamed as AWS20"                                                                                 |
| 86           | "a more compact and lower-power design"                                                                |
| 87           | "also use a R.M. Young"                                                                                |
| Fig 3        | Replace $T_{hut}$ with $T_{shield}$ on axis label. Replace "solar heating of the temperature hut" with |
|              | "solar heating of the temperature shield" in caption                                                   |
| 137          | "was measured inside a passively ventilated radiation shield, not in the ambient air, and              |
|              | could include radiative heating error."                                                                |
| 148          | "the same as is done"                                                                                  |
| 149          | "vapour pressures in the radiation shield"                                                             |
| 151          | "values <del>far a</del> bove 100%."                                                                   |
| 166          | "For each station we select"                                                                           |
| 170          | "Near-neutral conditions are defined as when wind speeds are higher than 6 ms-1,                       |
|              | relative humidity above 80%,"                                                                          |
| Fig. 5       | Shiftthis figureto after the "Pyranometer (short wave)" discussion                                     |
| 256 & 259    | Riming or hoar frost deposition? Or either?                                                            |
| Table 4      | Shift this Table to be with the text discussion of "Flagging"                                          |
| 277          | "The time-averages of all measurements"                                                                |
| Fig 8a capti | on T is green, not red                                                                                 |
| Fig 8a capti | on SW is orange, not yellow                                                                            |
| 321          | "effectiveness of the radiation shields."                                                              |
| 329          | "does not substantially differ from the reported"                                                      |
| 331          | "was never replaced after it's installation"                                                           |
| 340          | "less than 5% of the daily"                                                                            |
| 350          | "0.08 m <del>for b</del> etween the type I/II stations"                                                |
| 373          | This seems unnecessary as a reference as the url is also given                                         |

---

## Author Comment (AC1)

**Reply to reviewer comment on ESSDd preprint "IMAU Antarctic automatic weather station data, including surface radiation balance (1995–2022)"**
**https://doi.org/10.5194/essd-2025-88**

by Maurice van Tiggelen, Paul Smeets, Carleen Reijmer, Peter Kuipers Munneke and Michiel van den Broeke

We thank the editor and all three reviewers for the positive and constructive feedback regarding our dataset manuscript. Below we reply point-by-point to the editor comments and to all the three reviewer comments. The editor and reviewer comments are written in **black**, our reply is written in **blue** and our proposed changes are written in **orange**. We hope that the revised manuscript is suited for publication in ESSD.

On behalf of the co-authors,

Maurice van Tiggelen

**Editor Comment** on 15 April 2025

Figure 1: The use of the "rainbow" color map is generally discouraged in climate science, as it introduces artificial visual discontinuities, creates non-linear perceptual contrasts, and may mislead the interpretation of continuous gradients (see, e.g., Crameri et al., 2020). Moreover, it compromises accessibility, especially for individuals with visual impairments. Perceptually uniform color scales are strongly recommended for more accurate, inclusive, and accessible visual representation.

Crameri, F., Shephard, G.E., & Heron, P.J. (2020). The misuse of colour in science communication. Nature Communications, 11, 5444. https://doi.org/10.1038/s41467-020-19160-7

We have changed the colormap of Figure 1 accordingly. We have also included a scale for the wind vector, based on a suggestion by reviewer 2.

[Figure]

**Figure 1.** Location of the AWS presented in this database (red triangles). Background colour denotes the modelled annual average 2m near surface air temperature from regional climate model RACMO2.4p1 during the period 1990-2020 (Van Dalum et al, 2024). Black circles denote AWS from the AntAWS database (Wang et al, 2023). Average 10m near surface wind vectors from RACMO2.4p1 are also shown. Insets are shown for Dronning Maud Land (top left) and the Antarctic Peninsula (bottom left).

Line 65: There appears to be a discrepancy—if AWS18 is still operational, Table 1 should be updated accordingly. If not, it should read AWS20 for clarity.

This was indeed a typo that we have now corrected. We have also added AWS20 to Table 1, but we would like to point out that AWS20 is not part of this dataset. This is now also mentioned explicitly in the revised manuscript.

Line 65: The term "Type III" is used before it has been introduced. Consider either removing it or introducing Table 2 earlier in the manuscript.

We have adjusted this accordingly.

Typos:

Line 288: "sration"

Line 301: "both the two" → should be simplified to "both" or "the two".

**RC1**: 'Comment on essd-2025-88', Ian Allison, 07 May 2025

**General Comments**

This preprint provides an excellent description of more than 20 years of high quality and valuable Antarctic surface meteorological data from the IMAU automatic weather station (AWS) network. These AWS were designed to enable estimation of the ice sheet surface energy and mass balances. The publication clearly outlines what variables were measured, what instrumentation was used, how the data were processed and how corrections were made. The accuracy of the measurements is assessed in several ways including during periods when overlapping measurements from two AWS were made at five of the locations. The data availability from each station is summarised in both Figure and Table and the climatology at each site is also summarized.

Clear links are given to how the data can be accessed and to the software codes used to pre-process and correct the measurements. I particularly liked the simple flag assigned to each data sample alerting of potential problems in each of the measured variables.

The IMAU Antarctic AWS network is one of several that provide Antarctic surface meteorological data. Others include those of the University of Wisconsin, the Australian Antarctic Division and the Chinese Antarctic Programme. These are mentioned and acknowledged in the preprint. But I think that the larger Antarctic AWS data set can be more directly referenced with a few small changes that do not length the manuscript (the focus of which clearly should be the IMAU network). This could be simply done by noting in the Introduction that the locations of all AWS that have been deployed in Antarctica are shown on Figure 1 and that details of these stations are given in Wang et al., 2023 (reference already cited).

I could find no mention of what height the sensors were initially deployed at: this should be included around line 76. Also, while it is implicit that the wind, temperature and humidity data are corrected for height changes from snow accumulation/ablation, that is not explicitly noted until line 240. Perhaps a comment should be added in the introduction to the effect that "after correction for sensor errors and surface height change from snow accumulation/ablation, the

wind, temperature and humidity data are corrected to standard heights for SEB calculations using Monin-Obukhov similarity theory."

This manuscript clearly fits the objectives and standards of ESSD. I think it should be published after only minor changes and consideration of the following specific and technical comments.

We thank the reviewer for his time and valuable comments. We now refer to other AWS networks more directly in the introduction. We now also mention that the initial height of the sensors at installation varies between 2.6 m and 5 m above the surface, depending on the station and maintenance visit. We also now explicitly state in the introduction that the interpolated measurements at standard heights are also given in the dataset.

L29  **These AWS were part of one of several networks, such as the Antarctic Meteorological Research Center (AMRC) network maintained by the University of Wisonsin-Madison (Lazarra et al, 2012), the Australian Antarctic Division (AAD) network (Allison, 1998), the Italian National Program of Antarctic Research (PNRA) network (Grigioni et al, 2016), the Chinese National Antarctic Research Expedition (CHINARE) PANDA network (Ding et al, 2022), the British Antarctic Survey (BAS) network, the Japanese Antarctic Research Expedition (JARE) network (Kurita et al, 2024), the French Antarctic Program (Institut Polaire Francais-Paul Emile Victor, IPEV) network, or other similar networks maintained by different nations or organisations. These stations are shown in Figure 1, and further described in Wang et al. (2023).**

**Specific Comments (referenced to line number)**

68        A brief description here of what is done during maintenance visits would be useful.  Some details are given later in 2.4.  A problem we experienced with AWS high on the plateau is that the acoustic lens of the Cambel SR50 failed after a year or so in very cold conditions.  Was that a problem with IMAU AWS and, if so, was it fixed during maintenance?

We have added a small description of the work done during maintenance visits. Regarding the SR50 data, we have not experienced any problems at AWS09, AWS12 and AWS13, and the data is available for the entire record. We now explicitly mention this in the revised manuscript.

L83 **Maintenance visits were in general performed using a standard procedure contained in a form describing a list of actions, i.e. make photographs at**

**arrival, note anything unusual, measure yard directions and heights at arrival and departure, check datalogger data and replace the memory module. Sensors were commissioned to be replaced on a regular basis. Additional instructions and replacements were added in case of transmitted ARGOS data indicating failure of a sensor, of the datalogger, or of the power supply.**

L304 **At the plateau sites AWS12 and AWS13, no pressure data is available due to a malfunctioning datalogger, and no wind speed and wind direction data is available after 70 days of operation at AWS13 due to a malfunctioning sensor. Yet, despite the very cold and dry climate, most other variables including surface height are available for the entire period at AWS12 and AWS13.**

Table 2        I think that this table should be relocated after Section 2.2.  Reading Table 2 first I was left wondering what sort of radiation shields were used, whether the anemometer model numbers refer to propellor or vane anemometers, etc. These are addressed in the text.

We have moved Table 2 further down.

Table 2        What depth is snow temperature measured at? Is it also corrected for subsequent accumulation?

The snow temperature was measured at different depths that vary between stations and maintenance visits. Since the recorded snow temperature is not part of this dataset, we do not provide the exact list of depths. Providing this information would also require a careful analysis of historical field reports, which is outside the scope if this dataset. We now mention this in the manuscript.

L112 **At all locations, the AWS were also fitted with thermistor strings to measure the subsurface temperatures. However, these subsurface data are not part of this quality-controlled dataset, since the exact installation depth of the subsurface sensors is not known for all maintenance visits.**

100             Riming or hoar frost deposition? Inland station sensors are usually affected by hoar frost deposition (ice crystals formed by direct sublimation of water vapour) rather than rime (ice formed from freezing of supercooled liquid water). Coastal stations may suffer riming.

We agree that there is a clear distinction, but we can't separate these processes in a reliable way based on our measurements alone. We have thus rephrased this sentence.

L118 All unheated meteorological instruments operating in polar conditions may suffer from riming **or hoar frost deposition**, which we assume to occur when the relative humidity exceeds 90 %,  the air temperature is lower than  0 °C and the absolute value of the net longwave radiation is smaller than 2 Wm$^{-2}$ for at least 24 consecutive hours.

L276 A binary quality flag is generated for each sample that aims to incorporate all the possible combinations of suspicious or missing data for each measured variable and possible riming **or hoar frost deposition**, **hereafter denoted 'riming'**, […]

118        Does "hut temperature" mean "temperature inside the radiation shield"?  I think the use of "hut" is inappropriate.

We have replaced the term "hut temperature" by "temperature measured inside the radiation shield" in the revised manuscript.

153 & 186    Largest radiation measurement errors probably occur with hoar frost deposition on the instruments.  These events are flagged, but no corrections are made. An additional note regarding this might be added here.

We have added this information in the revied manuscript.

314 & 332    Were the sonic anemometers used at the AWS14 and AWS05 site comparisons separate instruments, not part of any AWS?

The reviewer is correct, these observations were from different experiments and are not part of this dataset. We now explicitly mention this in the revised manuscript.

**Technical and minor comments (referenced to line number)**

10        This seems unnecessary as a reference as it just refers to the url given in the preceding line (https://doi.pangaea.de/10.1594...............)

We note that this formatting is a requirement of ESSD. No modification done.

51        "International Polar Year (IPY 2007-2008)"

Adapted

54        "one AWS was installed at former"

Corrected

Fig 1        Location labelling of AWS 04, 05, 06, 07, 08, 09 is not clear in the main picture, but is clarified in the inset

We believe that the location of each AWS is sufficiently clear from the the insets and from the coordinates in Table 1, hence no modification was done.

58          "CIRES and the the Jet Propulsion Laboratory"

Corrected

59          "in SCAR Inlet"

Corrected

62          "the Belgian Royal Meteorological Institute" or "the Royal Meteorological Institute of Belgium"

Corrected

63          "Belgian Princess Elizabeth station"

Corrected

Table 1        "and renamed as AWS20"

Corrected

86          "a more compact and lower-power design"

Adapted

87          "also use a R.M. Young"

Corrected

Fig 3          Replace $T_{hut}$ with $T_{shield}$ on axis label.  Replace "solar heating of the temperature hut" with "solar heating of the temperature shield" in caption

Done

137          "was measured inside a passively ventilated radiation shield, not in the ambient air, and could include radiative heating error."

Adapted

148          "the same as is done"

Adapted

149       "vapour pressures in the radiation shield"

Adapted

151       "values far above 100%."

No change done

166       "For each station we select"

Adapted

170       "Near-neutral conditions are defined as when wind speeds are higher than 6 ms−1, relative humidity above 80%,"

Adapted

Fig. 5       Shift this figure to after the "Pyranometer (short wave)" discussion

Done

256 & 259    Riming or hoar frost deposition? Or either?

These can't be separated with these observations, hence we now refer to both.

Table 4       Shift this Table to be with the text discussion of "Flagging"

Done

277       "The time-averages of all measurements"

Corrected

Fig 8a caption   T is green, not red

Corrected

Fig 8a caption   SW is orange, not yellow

Corrected

321       "effectiveness of the radiation shields."

Adapted

329        "does not substantially differ from the reported"

Corrected

331        "was never replaced after it's installation"

Corrected

340        "less than 5% of the daily"

Corrected

350        "0.08 m for between the type I/II stations"

Corrected

373        This seems unnecessary as a reference as the url is also given

We note this is standard citation formatting for a dataset. No modification done.

**RC2**: 'Comment on essd-2025-88', Anonymous Referee #2, 13 May 2025

**Comments on "IMAU Antarctic automatic weather station data, including surface radiation balance (1995-2022)"**

Given the critical impact of the Antarctic ice sheet on global climate change and sea level rise, it is urgent to develop an improved in situ meteorological observation network. This manuscript reported almost 30 years of Antarctic surface meteorological observations from 19 Automated Weather Stations (AWS) operated by the Institute for Marine and Atmospheric research Utrecht (IMAU) at Utrecht University, including especially the surface energy and mass balance. The manuscript described in detail the variables, instrumentation, and processing of the observational data, and ultimately produced an accessible dataset of great importance to Antarctic climate change and surface mass balance studies. How the data were preprocessed and corrected were also described with detail in the manuscript. This is important for the assessment and development of regional climate models for the Antarctic Ice Sheet, the validation of remote sensing as well as reanalysis products, and contributes to an increased understanding of Antarctic surface climatology. The subject of this work is aligned with Earth System Science Data and is very valuable. I recommend publishing it in ESSD with minor revisions.

1. The introduction is written concisely and logically, but it is short enough to provide more background to the study. I would suggest that the authors give an appropriate description about the important role of Antarctica in the global climate system, as well as adding a more detailed overview of the meteorological

observation networks that are currently being set up. This could include more detail on some of the projects and programs already mentioned.

We thank the reviewer for these recommendations. The second was also made by the other two reviewers. We have added some sentences to highlight the importance of the Antarctic climate in general, and we now also more explicitly refer to the other AWS networks in the introduction.

L12 **The Antarctic ice sheet is the largest reservoir of freshwater, holding a global sea-level potential of 58 m (Morlighem et al, 2020), that also acts as a reliable record of the recent climate (e.g. EPICA, 2004). Due to its isolation, dry climate, and long austral winter, it also provides unique and often favourable locations for meteorological, astronomical, geophysical and upper atmosphere observations.**

2. Regarding the radiative component, some of the short-wave radiation in the previously released data (https://doi.pangaea.de/10.1594/PANGAEA.910473) was observed during the winter months (most likely during the polar night), and I am not sure whether this is due to instrumental error or some other phenomenon (may not be an error, but an objective phenomenon). I would like to know if this problem still exists in the range reported in this manuscript.

Erroneous non-zero shortwave radiation readings during nighttime or polar night are either due to the zero-offset from the pyranometer caused by longwave cooling of the radiometer body, or from a bias in the electronic sampling system. Both these biases are corrected for in this dataset, as described in L186-208 and shown in Figure 5, but were not corrected for in the dataset mentioned by the reviewer. We have now more explicitly mentioned this in the revised manuscript.

L47 **Although the measurements from this dataset are also partly contained in the datasets from Jakobs et al. (2020) and Wang et al. (2021), the data presented in this work have gone through an elaborate quality control and data correction strategy, which are specifically tailored for the unique combination of sensors and locations of the IMAU dataset.**

3. -Another similar problem is that some of the relative humidity in the previous data exceeded 100%. How this data was processed in this work?

The relative humidity is corrected for saturation with respect to ice and for the hygrometer sensitivity at low temperatures, but also for the difference between ambient temperature and temperature inside the radiation shield, as described in L133-151.  This correction procedure results in some relative humidity values exceeding 100%. Values exceeding 110% are flagged but not capped nor removed from the dataset, as mentioned in Table 4, such that a different humidity correction

can be applied by the user if absolutely required.  The correction was done differently in the other dataset mentioned by the referee in point 2.

 4. There are some acronyms that appear multiple times in the manuscripts, such as the British Antarctic Survey (BAS), and whether it is possible to retain them only in their first appearance.

We have made sure that each acronym is only defined once and then used in the remainder of the manuscript.

5. L66-67: Is it possible to correct for the effects of this positional movement on meteorological observations? This would be much more useful for the use of the data.

As far as we know, there is no benchmark or commonly used correction procedure for the lateral movement of meteorological observations over ice shelves. Furthermore, we do not think that such a correction is desirable for the published dataset, since this would involve a model for e.g. interpolation, thereby compromising the independence of these observations when used for model training or evaluation.

6. Figure 1: I think it would be useful to add a wind vector scale to measure wind speed.  -Similarly, it is suggested that climatological values for other atmospheric variables be given, which can add the reader's understanding of the Antarctic surface climatology. They can be placed in a supplementary file

We thank the reviewer for these suggestions. We have now added the wind vector scale to Figure 1. Regarding other maps of climatological variables, we prefer to refer to other recent work such as Van Dalum et al, 2024, in order to keep the focus of this manuscript to meteorological observations.

**RC3**: ['Comment on essd-2025-88'](), David Bromwich, 20 May 2025

General Comments:

This manuscript is a welcome addition that describes an important set of surface observations from automatic weather stations (AWS) across the data sparse Antarctic continent. The authors provide a comprehensive description of their extensive quality controls. Such care is needed with observations from AWS in remote locations that often experience data collection challenges (e.g., Lazzara et al. 2012). The radiation fluxes are a valuable addition to those provided by the four

BSRN stations in Antarctica. One could assume from the manuscript text that the IMAU Antarctic AWS program is mostly concluded. If so, this manuscript is an especially landmark effort.

Mostly my comments are centered around the context issue. The system requires me to rate this as major revisions, but it is more like minor with a few more important aspects.

1. Are any of these observations part of the AntAWS database? I presume not from Figure 1, but it is important to make this very clear. And this encourages the merging of these two data streams.

In fact, many IMAU stations from our dataset were also included in the AntAWS dataset. However, the AntAWS dataset does not include measured radiative components nor surface height, and the postprocessing including quality control, data corrections and data flagging were either not done or done differently. We have now explicitly mentioned this is the revised manuscript.

L47 **Although the measurements from this dataset are also partly contained in the datasets from Jakobs et al. (2020) and Wang et al. (2021), the data presented in this work have gone through an elaborate quality control and data correction strategy, which are specifically tailored for the unique combination of sensors and locations of the IMAU dataset.**

2. Were any of these observations made available to the GTS in near-realtime? If not, then the value of the observations increases because the reanalyses would not assimilate them, making the IMAU observations an independent test of the reanalyses.

No, these stations have not been assimilated in any reanalysis product. This is indeed important information that we have now added to the revised manuscript.

L113 **The measurements have never been transmitted to the Global Telecommunication System (GTS) and not been assimilated in reanalysis products.**

L396 **The measurements from these stations have not been assimilated, thereby allowing for an independent benchmarking of reanalysis products.**

3. This manuscript really underplays the AWS program run by the University of Wisconsin-Madison that forms the backbone of the current Antarctic AWS network, e.g., Lazzara et al. (2012). Please rectify.

We now more clearly refer to other AWS networks in the introduction of the revised manuscript, something that was also requested by the other 2 reviewers.

L29 **These AWS were part of one of several networks, such as the Antarctic Meteorological Research Center (AMRC) network maintained by the University of Wisonsin-Madison (Lazarra et al, 2012), the Australian Antarctic Division (AAD) network (Allison, 1998), the Italian National Program of Antarctic Research (PNRA) network (Grigioni et al, 2016), the Chinese National Antarctic Research Expedition (CHINARE) PANDA network (Ding et al, 2022), the British Antarctic Survey (BAS) network, the Japanese Antarctic Research Expedition (JARE) network (Kurita et al, 2024), the French Antarctic Program (Institut Polaire Francais-Paul Emile Victor, IPEV) network, or other similar networks maintained by different nations or organisations. These stations are shown in Figure 1, and further described in Wang et al. (2023).**

4. An important monthly data set of near-surface air temperature from AWS in eastern Queen Maud Land is Kurita et al. (2024, DOI: 10.1175/JTECH-D-23-0092.1). Please integrate into your manuscript.

We thank the referee for this suggestion and have now added this reference and some other relevant references to the introduction (reply just above).

5. I am particularly interested in the Kohnen site, AWS09, as a long-term observing site. Medley et al. (2017, https://doi.org/10.1002/2017GL075992) report 1.1C/decade warming at the Kohnen AWS, 1998-2016. I have doubts about this record because of the change in the temperature sensor in 2008 (mentioned here in Table 2) and the warming trend after this was much lower (Fig. S11). Here is the key passage from that manuscript: "Due to the low temperatures encountered at the site, some of the sensors operated outside their operational specifications. For this reason, the station was equipped with different temperature sensor in 2008." (from Section 2.2). Downloading the Kohnen 2-m corrected temperatures from Pangaea reveals almost completely missing air temperatures after 2010 whereas the uncorrected temperatures are present. Please discuss this situation in your manuscript.

We thank the reviewer for the interest and for taking the time to open the dataset. The temperature can only be interpolated to 2m height using similarity theory if all the variables are non-flagged and available, since the sensible heat flux must be computed using a surface energy balance model. At AWS09, the outgoing longwave radiation was missing due to a malfunctioning logger between 2009 and 2016. Then, from 2017 onwards, the relative humidity was missing due to a malfunctioning

sensor. This means that there is indeed almost no period with interpolated temperature data after 2009. On the other hand, the temperature at sensor height is available for the whole period. We now discuss this specific case at AWS09 is more detail in the revised manuscript.

L294 **At AWS09, the outgoing longwave radiation is missing between 2009 and 2016 due to a malfunctioning logger, while from 2017 onwards, the relative humidity is missing due to a malfunctioning sensor. This means that not more than one year of interpolated temperature data is available at AWS09 after 2009, since all variables need to be un-flagged to reliably interpolate quantities using Eqs. 9-11.**

6. David Bromwich May 20, 2025

---

## Author Response (AR2)

**Message to the editor regarding accepted ESSDd preprint "IMAU Antarctic automatic weather station data, including surface radiation balance (1995–2022)" https://doi.org/10.5194/essd-2025-88**

by Maurice van Tiggelen, Paul Smeets, Carleen Reijmer, Peter Kuipers Munneke and Michiel van den Broeke

We thank the editor for accepting our submission for final publication in ESSD. Based on a comment from reviewer 3 regarding whether the data was transmitted to the GTS, and possibly assimilated in re-analysis products, we have double-checked this to make sure, and found that in fact some raw, uncorrected data was transmitted to the GTS via Argos for some stations and some periods. This means that some specific AWS data from our network could have been assimilated in some reanalysis products, in contradiction to what was mentioned in the previous version of the manuscript.

We now corrected this in the revised manuscript at L116 and L405, and added in Table 1 for which stations and which periods this happened. It must be noted that this does not change the dataset in any way, but this information may be relevant for specific applications.

On behalf of the co-authors,

Maurice van Tiggelen

L116
Measurements of stations 5, 6, 9, 12, 13, 14, 15 and 17 have for a period of their operation been transmitted to the Global Telecommunication System (GTS). The exact periods are given in Table 1.

L406
Some of the measurements were transmitted to the GTS system and are therefore available to be assimilated in reanalyses products. The exact periods and stations for which this is the case are marked in Table 1. The other measurements have not been assimilated and thereby allow for an independent benchmark for reanalyses products.